# FLOW STRAIGHT AND FAST: LEARNING TO GENERATE AND TRANSFER DATA WITH RECTIFIED FLOW

**Xingchao Liu**[∗]**, Chengyue Gong,**[∗] **Qiang Liu**
Department of Computer Science
University of Texas at Austin
`{xcliu, cygong, lqiang}@cs.utexas.edu`

## ABSTRACT

We present rectified flow, a simple approach to learning (neural) ordinary differential equation (ODE) models to transport between two empirically observed distributions $\pi_0$ and $\pi_1$, hence providing a unified solution to generative modeling and domain transfer, among various other tasks involving distribution transport. The idea of rectified flow is to learn the ODE to follow the straight paths connecting the points drawn from $\pi_0$ and $\pi_1$ as much as possible. This is achieved by solving a straightforward nonlinear least squares optimization problem, which can be easily scaled to large models without introducing extra parameters beyond standard supervised learning. The straight paths are the shortest paths between two points, and can be simulated exactly without time discretization and hence yield computationally efficient models. We show that, by learning a rectified flow from data, we effectively turn an arbitrary coupling of $\pi_0$ and $\pi_1$ to a new deterministic coupling with provably non-increasing convex transport costs. In addition, with a "reflow" procedure that iteratively learns a new rectified flow from the data bootstrapped from the previous one, we obtain a sequence of flows with increasingly straight paths, which can be simulated accurately with coarse time discretization in the inference phase. In empirical studies, we show that rectified flow performs superbly on image generation and image-to-image translation. In particular, on image generation and translation, our method yields nearly straight flows that give high quality results even with *a single Euler discretization step*. Code is available at https://github.com/gnobitab/RectifiedFlow.

## 1 INTRODUCTION

Compared with supervised learning, the shared difficulty of various forms of unsupervised learning is the lack of *paired* input/output data that makes standard regression or classification tasks possible. The crux of many unsupervised methods is to find meaningful correspondences between points from two distributions. For example, generative models such as generative adversarial networks (GAN) and variational autoencoders (VAE) (e.g., Goodfellow et al., 2014; Kingma & Welling, 2013; Dinh et al., 2016) seek to map data points to latent codes following a simple elementary (e.g., Gaussian) distribution with which the data can be generated and manipulated. On the other hand, domain transfer methods find mappings to transfer points between two different data distributions, both observed empirically, for the purpose of image-to-image translation, style transfer, and domain adaption (e.g., Zhu et al., 2017; Flamary et al., 2016; Trigila & Tabak, 2016; Peyré et al., 2019). These tasks can be framed unifiedly as finding a transport map between two distributions:

**Learning Transport Mapping** *Given empirical observations of two distributions $\pi_0, \pi_1$ on $\mathbb{R}^d$, find a transport map $T\colon \mathbb{R}^d \to \mathbb{R}^d$, which, in the infinite data limit, gives $Z_1 := T(Z_0) \sim \pi_1$ when $Z_0 \sim \pi_0$, that is, $(Z_0, Z_1)$ is a coupling (a.k.a transport plan) of $\pi_0$ and $\pi_1$.*

We should note that the answers of this problem are not unique because there are often infinitely many transport maps between two distributions. *Optimal transport* (OT) (e.g., Villani, 2021; Ambrosio et al., 2021; Figalli & Glaudo, 2021; Peyré et al., 2019) addresses the more challenging

---

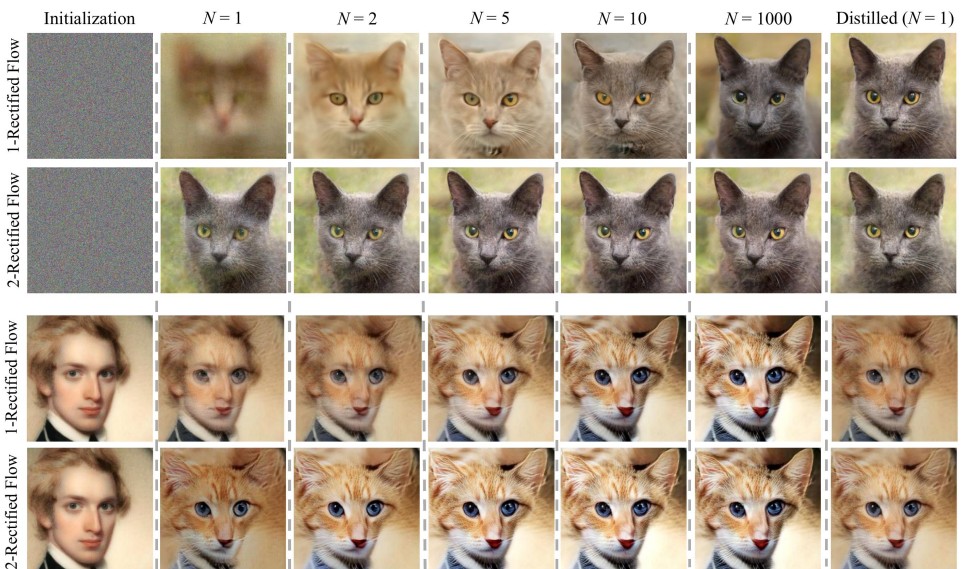

Figure 1: The trajectories of rectified flows for image generation ($\pi_0$: standard Gaussian noise, $\pi_1$: cat faces, top two rows), and image transfer between human and cat faces ($\pi_0$: human faces, $\pi_1$: cat faces, bottom two rows), when simulated using Euler method with step size $1/N$ for $N$ steps. The first rectified flow induced from the training data (*1-rectified flow*) yields good results with a very small number (e.g., $\geq 2$) of steps; the straightened reflow induced from *1-rectified flow* (denoted as *2-rectified flow*) has nearly straight line trajectories and yield good results even with one discretization step.

problem of finding an *optimal* coupling that minimizes a notion of transport cost, typically of form $\mathbb{E}[c(Z_1 - Z_0)]$, where $c: \mathbb{R}^d \to \mathbb{R}$ is a cost function, such as $c(x) = \|x\|^2$. However, for the generative and transfer modeling tasks above, the transport cost is not of direct interest, even though it induces a number of desirable properties. Hence, it is not necessary to accurate solve the OT problems given the high difficulty of doing so. An important question is to identify relaxed notions of optimality that are of direct interest for ML tasks and are easier to enforce in practice.

Several lines of techniques have been developed depending on how to represent and train the map $T$. In traditional generative models, $T$ is parameterized as a neural network, and trained with either GAN-type minimax algorithms or (approximate) maximum likelihood estimation (MLE). However, GANs suffer from numerically instability and mode collapse issues, and require substantial engineering efforts and human tuning, which tend to transfer poorly across different model architecture and datasets. On the other hand, MLE tends to be intractable for complex models, and hence requires either approximate (variational or Monte Carlo) inference techniques such as those used in VAE, or special model structures that yield tractable likelihood such as normalizing flow and auto-regressive models, which causes difficult trade-offs between expressive power and computational cost.

Recently, advances have been made by representing the transport plan *implicitly as a continuous time process*, including flow models with neural ordinary differential equations (ODEs) (e.g., Chen et al., 2018; Papamakarios et al., 2021; Song et al., 2020a) and diffusion models by stochastic differential equations (SDEs) (e.g., Song et al., 2020b; Ho et al., 2020; Tzen & Raginsky, 2019; De Bortoli et al., 2021; Vargas et al., 2021). In these models, a neural network is trained to represent the drift force of the processes and a numerical ODE/SDE solver is used to simulate the process during inference. By leveraging the mathematical structures of ODEs/SDEs, the continuous-time models can be trained efficiently without resorting to minimax or traditional approximate inference techniques. The most notable examples are the score-based generative models (Song & Ermon, 2019; 2020; Song et al., 2020b) and denoising diffusion probabilistic models (DDPM) (Ho et al., 2020), which has achieved impressive empirical results on image generation recently (e.g., Dhariwal & Nichol, 2021). However, compared with the traditional "one-step" models like GAN and VAE, continuous-times models are effectively "infinite-step" and cast high computational cost in inference time: drawing a single point (e.g., an image) requires to solve the ODE/SDE with a numerical solver that needs to repeatedly call the expensive neural force field for a large number of times.

Moreover, in existing approaches, generative modeling and domain transfer are typically treated separately. It often requires to extend techniques to solve domain transfer problems; see *e.g.*, Cycle

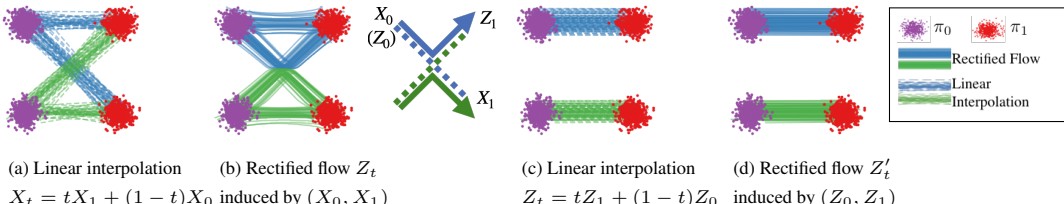

(a) Linear interpolation

$X_t = tX_1 + (1-t)X_0$ induced by $(X_0, X_1)$

(b) Rectified flow $Z_t$

(c) Linear interpolation

$Z_t = tZ_1 + (1-t)Z_0$ induced by $(Z_0, Z_1)$

(d) Rectified flow $Z_t'$

Figure 2: (a) Linear interpolation of data input $(X_0, X_1) \sim \pi_0 \times \pi_1$. (b) The trained rectified flow $Z_t$; the trajectories are "rewired" at the intersection points to avoid crossing. (c) The linear interpolation of the end points $(Z_0, Z_1)$ of flow $Z_t$. (d) The rectified flow induced from $(Z_0, Z_1)$, which follows straight paths.

GAN (Zhu et al., 2017) and diffusion-based image-to-image translation (e.g., Su et al., 2022; Zhao et al., 2022). One framework that naturally unifies both tasks is optimal transport, which, however, is challenging to solve for problems with high dimensional and large volumes of data.

**Contribution**  We introduce *rectified flow*, a surprisingly simple approach to the transport mapping problem, which unifiedly solves both generative modeling and domain transfer. The rectified flow is an ODE model that transport distribution $\pi_0$ to $\pi_1$ by *following straight line paths as much as possible*. The straight paths are preferred both theoretically because it is the shortest path between two end points, and computationally because it can be exactly simulated without time discretization. Hence, flows with straight paths bridge the gap between one-step and continuous-time models.

Algorithmically, the rectified flow is trained with a simple and scalable unconstrained least squares optimization procedure, which avoids the instability issues of GANs, the intractable likelihood of MLE methods, and the subtle hyper-parameter decisions of denoising diffusion models. The procedure of obtaining the rectified flow from the training data has the attractive theoretical property of 1) yielding a coupling with non-increasing transport cost jointly for all convex cost $c$, and 2) making the paths of flow increasingly straight and hence incurring lower error with numerical solvers. With a *reflow* procedure that iteratively trains new rectified flows with the data simulated from the previously obtained rectified flow, we obtain nearly straight flows that yield good results even with very coarse time discretization. Our method is purely ODE-based, both conceptually simpler and practically faster in inference time than the SDE-based methods (Ho et al., 2020; Song et al., 2020b;a).

As shown in Figure 1, rectified flow yields high-quality results for image generation when simulated with a very few number of Euler steps (see Figure 1, top row). Moreover, with just one step of reflow, the flow becomes nearly straight and hence yields good results with one Euler discretization step (Figure 1, the second row). This substantially improves over the standard denoising diffusion methods. Quantitatively, we claim a state-of-the-art result of FID (4.85) and recall (0.51) on CI-FAR10 for one-step fast diffusion/flow models (Bao et al., 2022; Lyu et al., 2022; Xiao et al., 2021; Zheng et al., 2022; Luhman & Luhman, 2021). The same algorithm also achieves superb results on the image-to-image translation task (see the bottom two rows of Figure 1).

## 2 METHOD

We provide an overview of the method in Section 2.1 with more discussion in Section 2.2.

### 2.1 OVERVIEW

**Rectified flow**  Given empirical observations of $X_0 \sim \pi_0$ and $X_1 \sim \pi_1$, the rectified flow induced from $(X_0, X_1)$ is an ordinary differentiable model (ODE) on time $t \in [0, 1]$,

$$\mathrm{d}Z_t = v(Z_t, t)\mathrm{d}t,$$

which converts $Z_0$ from $\pi_0$ to a $Z_1$ following $\pi_1$. The drift force $v \colon \mathbb{R}^d \to \mathbb{R}^d$ is set to drive the flow to follow the direction $(X_1 - X_0)$ of the linear path pointing from $X_0$ to $X_1$ as much as possible, by solving a simple least squares regression problem:

$$\min_v \int_0^1 \mathbb{E}\left[\left\|(X_1 - X_0) - v(X_t, t)\right\|^2\right] \mathrm{d}t, \quad \text{with} \quad X_t = tX_1 + (1-t)X_0, \tag{1}$$

---
**Algorithm 1** Rectified Flow: Main Algorithm

---
**Procedure**: $\boldsymbol{Z} = \texttt{RectFlow}((X_0, X_1))$:

  *Inputs*: Velocity model $v_\theta \colon \mathbb{R}^d \to \mathbb{R}^d$ with parameter $\theta$.

  *Training*: $\hat{\theta} = \underset{\theta}{\arg\min} \, \mathbb{E}\left[\|X_1 - X_0 - v(tX_1 + (1 - t)X_0, \, t)\|^2\right]$, with $t \sim \text{Uniform}([0, 1])$.

  *Sampling*: Draw $(Z_0, Z_1)$ following $\mathrm{d}Z_t = v_{\hat{\theta}}(Z_t, t)\mathrm{d}t$ starting from $Z_0 \sim \pi_0$ (or $Z_1 \sim \pi_1$).
  *Return*: $\boldsymbol{Z} = \{Z_t \colon t \in [0, 1]\}$.

  **Reflow** (optional): $\boldsymbol{Z}^{k+1} = \texttt{RectFlow}((Z_0^k, Z_1^k))$, starting from $(Z_0^0, Z_1^0) = (X_0, X_1)$, where $(X_0, X_1)$ is drawn from $\pi_0$ and $\pi_1$.

  **Distill** (optional): Learn a neural network $\hat{T}$ to distill the $k$-rectified flow, such that $Z_1^k \approx \hat{T}(Z_0^k)$.

---

where $X_t$ is the linear interpolation of $X_0$ and $X_1$. The expectation $\mathbb{E}[\cdot]$ here is w.r.t. the randomness of $(X_0, X_1)$ while treating $X_t = tX_1 + (1 - t)X_0$ as a function of $(X_0, X_1)$. To understand the method intuitively, note that the linear interpolation $X_t$ follows an naive ODE of $\mathrm{d}X_t = (X_1 - X_0)\mathrm{d}t$. This ODE is not practically useful for constructing a transport map as it is *non-causal* (or anticipating): the update of $X_t$ requires the information of the final point $X_1$. By fitting the drift $v$ with $X_1 - X_0$, the rectified flow *causalizes* the paths of linear interpolation $X_t$, yielding an ODE flow that can be simulated without seeing the future.

In practice, we parameterize $v$ with neural networks or other nonlinear models and solve (1) with any off-the-shelf stochastic optimizer, such as stochastic gradient descent, with empirical draws of $(X_0, X_1)$. After we get $v$, we solve the ODE either forwardly starting from $Z_0 \sim \pi_0$ to transfer $\pi_0$ to $\pi_1$, or backwardly from $Z_1 \sim \pi_1$ to $Z_0 \sim \pi_0$. Specifically, for backward sampling, we simply solve $\mathrm{d}\tilde{X}_t = -v(\tilde{X}_t, t)\mathrm{d}t$ initialized from $\tilde{X}_0 \sim \pi_1$ and set $X_t = \tilde{X}_{1-t}$. The forward and backward sampling are equally favored by the training method, because the loss in (1) is *time-symmetric* in that it yields the equivalent problem when exchanging $X_0$ and $X_1$ and flipping the sign of $v$.

**Flows avoid crossing**  A key to understanding the method is the non-crossing property of flows as illustrated in Figure 2: the different paths of a well-defined ODE $\mathrm{d}Z_t = v(Z_t, t)\mathrm{d}t$, whose solution exists and is unique, *cannot cross each other* at any time $t \in [0, 1)$. Specifically, there exists no location $z \in \mathbb{R}^d$ and time $t \in [0, 1)$, such that two paths go across $z$ at time $t$ along different directions, because otherwise the solution of the ODE would be non-unique. On the other hand, the paths of the interpolation process $X_t$ may intersect with each other (Figure 2a), which what makes it non-causal. Hence, as shown in Figure 2b, by solving the optimization in (1), the rectified flow *rewires* the individual trajectories passing through the intersection points to avoid crossing, while tracing out the same density map as that of the linear interpolation paths. We can view the linear interpolation $X_t$ as building roads (or tunnels) that connect $\pi_0$ and $\pi_1$, and the rectified flow as traffics of particles passing through the roads in a myopic, memoryless, non-crossing way, which ignore the global path information of how $X_0$ and $X_1$ are paired, and rebuild a more deterministic pairing of $(Z_0, Z_1)$.

**Rectified flows reduce transport costs**  If (1) is solved exactly, the pair $(Z_0, Z_1)$ of the rectified flow is guaranteed to be a valid coupling of $\pi_0, \pi_1$ (Theorem D.3), that is, $Z_1$ follows $\pi_1$ if $Z_0 \sim \pi_0$. Moreover, $(Z_0, Z_1)$ guarantees to yield no larger transport cost than the data pair $(X_0, X_1)$ simultaneously for *all* convex cost functions $c$ (Theorem D.5). The data pair $(X_0, X_1)$ can be an arbitrary coupling of $\pi_0, \pi_1$, typically independent (i.e., $(X_0, X_1) \sim \pi_0 \times \pi_1$) as dictated by the lack of meaningfully paired observations in practical problems. In comparison, the rectified coupling $(Z_0, Z_1)$ has a deterministic dependency as it is constructed from an ODE model. Denote by $(Z_0, Z_1) = \texttt{RectFlow}((X_0, X_1))$ the "rectification" map from $(X_0, X_1)$ to $(Z_0, Z_1)$. Hence, $\texttt{RectFlow}(\cdot)$ converts an arbitrary coupling into a deterministic coupling with lower convex transport costs.

**Straight line flows yield fast simulation**  Denote by $\boldsymbol{Z} = \texttt{RectFlow}((X_0, X_1))$ the rectified flow induced from $(X_0, X_1)$. Applying this operator recursively following Algorithm 1 yields a sequence of rectified flows $\boldsymbol{Z}^{k+1} = \texttt{RectFlow}((Z_0^k, Z_1^k))$ with $(Z_0^0, Z_1^0) = (X_0, X_1)$, where $\boldsymbol{Z}^k$ is the $k$-th rectified flow, or simply $k$-*rectified flow*, induced from $(X_0, X_1)$.

This *reflow* procedure not only decreases transport cost, but also has the important effect of straightening the paths of rectified flows, that is, making the paths of the flow more straight. This is highly

attractive computationally as flows with nearly straight paths incur small time-discretization error in numerical simulation. Indeed, perfectly straight paths can be simulated *exactly* with a single Euler step and is effectively a *one-step* model. This addresses the very bottleneck of high inference cost in existing continuous-time ODE/SDE models.

## 2.2 MAIN PROPERTIES

We provide more in-depth discussions on the main properties of rectified flow. We highlight the intuitions in this section and defer the full course theoretical analysis to Appendix D.

First, for a given input coupling $(X_0, X_1)$, the exact minimum of (1) is achieved if

$$v^X(x, t) = \mathbb{E}[X_1 - X_0 \mid X_t = x], \tag{2}$$

which is the expectation of the line directions $X_1 - X_0$ that pass through $x$ at time $t$. We discuss below the property of rectified flow $\mathrm{d}Z_t = v^X(Z_t, t)\mathrm{d}t$ with $Z_0 \sim \pi_0$, assuming that the ODE has an unique solution.

**Marginal preserving property** [Theorem D.3]  *The pair $(Z_0, Z_1)$ is a coupling of $\pi_0$ and $\pi_1$. The marginal law of $Z_t$ equals that of $X_t$ at every time $t$, that is,* $\mathrm{Law}(Z_t) = \mathrm{Law}(X_t), \forall t \in [0, 1]$.

Intuitively, this holds because, by the definition of $v^X$ in (2), the expected amount of mass that passes through every infinitesmal volume at all location and time are equal under the dynamics of $X_t$ and $Z_t$, which ensures that they trace out the same marginal distributions:

$$\text{Flow in \& out} \left( \begin{array}{c} \includegraphics \end{array} \right) = \text{Flow in \& out} \left( \begin{array}{c} \includegraphics \end{array} \right), \forall \text{time \& location} \Rightarrow \mathrm{Law}(Z_t) = \mathrm{Law}(X_t), \forall t.$$

Mathematically, this results is proved in Section D.1 by showing that the marginal laws of both $X_t$ and $Z_t$ satisfy the same continuity equation $\dot{\rho}_t + \nabla \cdot (v^X \rho_t) = 0$, and are hence equivalent.

On the other hand, the joint distributions of the whole trajectory of $Z_t$ and that of $X_t$ are different in general. In particular, $X_t$ is in general a non-causal, non-Markov process, with $(X_0, X_1)$ a stochastic coupling, and $Z_t$ *causalizes*, *Markovianizes* and *derandomizes* $X_t$, while preserving the marginal distributions at all time.

**Reducing transport costs** [Theorem D.5]  *The coupling $(Z_0, Z_1)$ yields no larger convex transport costs than the input $(X_0, X_1)$ in that $\mathbb{E}[c(Z_1 - Z_0)] \leq \mathbb{E}[c(X_1 - X_0)]$ for any convex $c \colon \mathbb{R}^d \to \mathbb{R}$.*

The transport costs measure the expense of transporting the mass of one distribution to another following the assignment relation specified by the coupling and is a central topic in optimal transport (e.g., Villani, 2009; 2021; Santambrogio, 2015; Peyré et al., 2019; Figalli & Glaudo, 2021). Typical examples are $c(\cdot) = \|\cdot\|^\alpha$ with $\alpha \geq 1$. Hence, `RectFlow`$(\cdot)$ yields a Pareto descent on the collection of all convex transport costs, without targeting any specific $c$. This distinguishes it from the typical optimal transport optimization methods, which are explicitly framed to optimize a given $c$. As a result, recursive application of `RectFlow`$(\cdot)$ does not guarantee to attain the $c$-optimal coupling for any given $c$, with the exception in the one-dimensional case when the fixed point of `RectFlow`$(\cdot)$ coincides with the unique monotonic coupling that simultaneously minimizes all non-negative convex costs $c$; see Appendix D.4.

Intuitively, the convex transport costs are guaranteed to decrease because the paths of the flow $Z_t$ is a rewiring of the straight paths connecting $(X_0, X_1)$. To give an illustration, consider the simple case of $c(\cdot) = \|\cdot\|$ when transport costs $\mathbb{E}[\|X_0 - X_1\|]$ and $\mathbb{E}[\|Z_0 - Z_1\|]$ are the expected length of the straight lines connecting the end points. The inequality can be proved graphically as follows:

$$\mathbb{E}[\|Z_0 - Z_1\|] = \text{Length}\left(\begin{array}{c}\includegraphics\end{array}\right) \overset{(*)}{\leq} \text{Length}\left(\begin{array}{c}\includegraphics\end{array}\right) \overset{(**)}{=} \text{Length}\left(\begin{array}{c}\includegraphics\end{array}\right) = \mathbb{E}[\|X_0 - X_1\|],$$

where $\overset{(*)}{\leq}$ uses the triangle inequality, and $\overset{(**)}{=}$ holds because the paths of $Z_t$ is a rewiring of the straight paths of $X_t$, following the construction of $v^X$ in (2). For general convex $c$, a similar proof using Jensen's inequality is shown in Appendix D.2.

**Reflow, straightening, fast simulation**  As shown in Figure 3, when we recursively apply the procedure $\boldsymbol{Z}^{k+1} = \texttt{RectFlow}((Z_0^k, Z_1^k))$, the paths of the $k$-rectified flow $\boldsymbol{Z}^k$ are increasingly

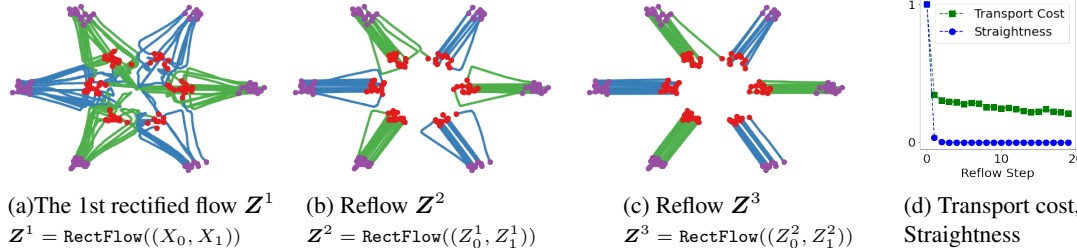

(a) The 1st rectified flow $\boldsymbol{Z}^1$
$\boldsymbol{Z}^1 = \texttt{RectFlow}((X_0, X_1))$

(b) Reflow $\boldsymbol{Z}^2$
$\boldsymbol{Z}^2 = \texttt{RectFlow}((Z_0^1, Z_1^1))$

(c) Reflow $\boldsymbol{Z}^3$
$\boldsymbol{Z}^3 = \texttt{RectFlow}((Z_0^2, Z_1^2))$

(d) Transport cost, Straightness

Figure 3: (a)-(c) Trajectories of the reflows on a toy example ($\pi_0$: purple dots, $\pi_1$: red dots; the green and blue lines are trajectories connecting different modes of $\pi_0, \pi_1$). (d) The straightness and the relative L2 transport cost v.s. the reflow steps. See Appendix D.5 for more information.

straight, and hence easier to simulate numerically, as $k$ increases. This straightening tendency can be guaranteed theoretically.

Specifically, we say that a flow $\mathrm{d}Z_t = v(Z_t, t)\mathrm{d}t$ is straight if we have almost surely that $Z_t = tZ_1 + (1-t)Z_0$ for $\forall t \in [0,1]$, or equivalently $v(Z_t, t) = Z_1 - Z_0 = const$ following each path. (More precisely, "straight" here refers to straight with a constant speed.) Such straight flows are highly attractive computationally as it is effective a one-step model: a single Euler step update $Z_1 = Z_0 + v(Z_0, 0)$ calculates the exact $Z_1$ from $Z_0$. Note that the linear interpolation $\boldsymbol{X} = \{X_t\}$ is straight by this definition but it is not a (causal) flow and hence can not be simulated without an oracle assess to draws of both $\pi_0$ and $\pi_1$. In comparison, it is non-trivial to make a flow $\mathrm{d}Z_t = v(Z_t, t)\mathrm{d}t$ straight, because if so $v$ must satisfy the inviscid Burgers' equation $\partial_t v + (\partial_z v)v = 0$:

$$\frac{\mathrm{d}}{\mathrm{d}t} v(Z_t, t) = \partial_z v(Z_t, t)\dot{Z}_t + \partial_t v(Z_t, t) = \partial_z v(Z_t, t)v(Z_t, t) + \partial_t v(Z_t, t) = 0.$$

More generally, the straightness of any smooth process $\boldsymbol{Z} = \{Z_t\}$ can be measured by

$$S(\boldsymbol{Z}) = \int_0^1 \mathbb{E}\left[\left\|(Z_1 - Z_0) - \dot{Z}_t\right\|^2\right] \mathrm{d}t. \tag{3}$$

$S(\boldsymbol{Z}) = 0$ means exact straightness. A flow with a small $S(\boldsymbol{Z})$ has nearly straight paths and hence can be simulated accurately using numerical solvers with a small number of discretization steps. Appendix D.3 shows that applying rectification recursively provably decreases $S(\boldsymbol{Z})$ towards zero.

[Theorem D.7] Let $\boldsymbol{Z}^k$ be the $k$-th rectified flow induced from $(X_0, X_1)$. Then $\min_{k \in \{0 \cdots K\}} S(\boldsymbol{Z}^k) \leq \frac{\mathbb{E}[\|X_1 - X_0\|^2]}{K}$.

As shown Figure 1, applying one step of reflow can already provide nearly straight flows that yield good performance when simulated with a single Euler step. It is not recommended to apply too many reflow steps as it may accumulate estimation error on $v^X$.

**Distillation**    After obtaining the $k$-th rectified flow $\boldsymbol{Z}^k$, we can further improve the inference speed by distilling the relation of $(Z_0^k, Z_1^k)$ into a neural network $\hat{T}$ to directly predict $Z_1^k$ from $Z_0^k$ without simulating the flow. Given that the flow is already nearly straight (and hence well approximated by the one-step update), the distillation can be done efficiently. In particular, if we take $\hat{T}(z_0) = z_0 + v(z_0, 0)$, then the loss function for distilling $\boldsymbol{Z}^k$ is $\mathbb{E}\left[\left\|(Z_1^k - Z_0^k) - v(Z_0^k, 0)\right\|^2\right]$, which is the term in (1) when $t = 0$. The difference between distillation and rectification should be highlighted: distillation attempts to faithfully approximate the coupling $(Z_0^k, Z_1^k)$, while rectification yields a different coupling $(Z_0^{k+1}, Z_1^{k+1})$ with lower transport cost and more straight flow. Hence, distillation should be applied only in the final stage for fine-tuning the model to one-step inference.

**Nonlinear rectified flow and friends** (Appendix C)    Our method can be extended to a simple yet high general *nonlinear rectified flow* method which uses *any smooth interpolation curve $X_t$* between $X_0$ and $X_1$ and a training loss of $\int_0^1 \mathbb{E}[\|\dot{X}_t - v(X_t, t)\|^2]\mathrm{d}t$, where $\dot{X}_t = \partial X_t$ is the time derivative. A natural class of $X_t$, which include probability flow ODEs (Song et al., 2020b) and DDIM (Song et al., 2020a) as the special cases, is $X_t = \alpha_t X_1 + \beta_t X_0$ for some $\alpha_t, \beta_t$ sequences satisfies (approximately) $\alpha_1 = \beta_0 = 1$ and $\alpha_0 = \beta_1 = 0$. However, depending on the choices of $\alpha_t, \beta_t$, the $X_t$ may not have straight trajectories and can not yield straight flows even with reflows.

Interestingly, the idea of learning generative ODEs with interpolating curves (e.g., 1-rectified flow with general $X_t$) was concurrently proposed in a number of works (Lipman et al., 2022; Albergo &

| Method | NFE(↓) | IS (↑) | FID (↓) | Recall (↑) |
|---|---|---|---|---|
| *ODE* | *One-Step Generation (Euler solver, N=1)* | | | |
| **1-Rectified Flow (+Distill)** | 1 | 1.13 (**9.08**) | 378 (*6.18*) | 0.0 (*0.45*) |
| **2-Rectified Flow (+Distill)** | 1 | 8.08 (*9.01*) | 12.21 (**4.85**) | 0.34 (*0.50*) |
| **3-Rectified Flow (+Distill)** | 1 | 8.47 (*8.79*) | 8.15 (*5.21*) | 0.41 (**0.51**) |
| VP ODE (Song et al., 2020b) (+*Distill*) | 1 | 1.20 (*8.73*) | 451 (*16.23*) | 0.0 (*0.29*) |
| sub-VP ODE (Song et al., 2020b) (+*Distill*) | 1 | 1.21 (*8.80*) | 451 (*14.32*) | 0.0 (*0.35*) |
| *ODE* | *Full Simulation (Runge–Kutta (RK45), Adaptive N)* | | | |
| **1-Rectified Flow** | 127 | **9.60** | **2.58** | **0.57** |
| **2-Rectified Flow** | 110 | 9.24 | 3.36 | 0.54 |
| **3-Rectified Flow** | 104 | 9.01 | 3.96 | 0.53 |
| VP ODE (Song et al., 2020b) | 140 | 9.37 | 3.93 | 0.51 |
| sub-VP ODE (Song et al., 2020b) | 146 | 9.46 | 3.16 | 0.55 |
| *SDE* | *Full Simulation (Euler solver, N=2000)* | | | |
| VP SDE (Song et al., 2020b) | 2000 | 9.58 | 2.55 | 0.58 |
| sub-VP SDE (Song et al., 2020b) | 2000 | 9.56 | 2.61 | 0.58 |

(a) Results using the DDPM++ architecture.

| Method | NFE(↓) | IS (↑) | FID (↓) | Recall (↑) |
|---|---|---|---|---|
| *GAN* | *One-Step Generation* | | | |
| SNGAN (Miyato et al., 2018) | 1 | 8.22 | 21.7 | 0.44 |
| StyleGAN2 (Karras et al., 2020) | 1 | 9.18 | 8.32 | 0.41 |
| StyleGAN-XL (Sauer et al., 2022) | 1 | - | 1.85 | 0.47 |
| StyleGAN2 + ADA (Karras et al., 2020) | 1 | 9.40 | 2.92 | 0.49 |
| StyleGAN2 + DiffAug (Zhao et al., 2020) | 1 | 9.40 | 5.79 | 0.42 |
| TransGAN + DiffAug (Jiang et al., 2021) | 1 | 9.02 | 9.26 | 0.41 |
| *GAN with U-Net* | *One-step Generation* | | | |
| TDPM (T=1) (Zheng et al., 2022) | 1 | 8.65 | 8.91 | 0.46 |
| Denoising Diffusion GAN (T=1) (Xiao et al., 2021) | 1 | 8.93 | 14.6 | 0.19 |
| *ODE* | *One Step Generation (Euler solver, N=1)* | | | |
| DDIM Distillation (Luhman & Luhman, 2021) | 1 | 8.36 | 9.36 | 0.51 |
| NCSN++ (VE ODE) (Song et al., 2020b) (+*Distill*) | 1 | 1.18 (2.57) | 461 (254) | 0.0 (0.0) |
| Progressive (Salimans & Ho, 2021) | 1 | - | 9.12 | - |
| DDIM (Song et al., 2020a) | 1 | - | >20 | - |
| *ODE* | *Full Simulation (Runge-Kutta (RK45), Adaptive N)* | | | |
| NCSN++ (VE ODE) (Song et al., 2020b) | 176 | 9.35 | 5.38 | 0.56 |
| *SDE* | *Full Simulation (Euler solver)* | | | |
| DDPM (Ho et al., 2020) | 1000 | 9.46 | 3.21 | 0.57 |
| NCSN++ (VE SDE) (Song et al., 2020b) | 2000 | 9.83 | 2.38 | 0.59 |
| *ODE* | *Full Simulation (Euler solver)* | | | |
| DDIM (Song et al., 2020a) | 10 | - | 13.36 | - |
| DDIM (Song et al., 2020a) | 100 | - | 4.16 | - |

(b) Recent results with different architectures reported in literature.

Table 1: Results on CIFAR10 unconditioned image generation. Fréchet Inception Distance (FID) and Inception Score (IS) measure the quality of the generated images, and recall score (Kynkäänniemi et al., 2019) measures diversity. The number of function evaluation (NFE) denotes the number of times we forward through the main neural network. It coincides with the number of discretization steps $N$ for ODE and SDE models.

Vanden-Eijnden, 2022; Heitz et al., 2023). Neklyudov et al. (2022) proposes a different but highly related method for the problem of trajectory inference from uncorrelated samples (e.g., Hashimoto et al., 2016; Lavenant et al., 2021). The distinctive features of our work include enabling fast, one-step generation via ODEs with straight trajectories using the *reflow* procedure, and the clarified connections to optimal transport (OT) shown in Appendix D, which is further elaborated in a companion work (Liu, 2022a). A different connection to OT was discussed in Albergo & Vanden-Eijnden (2022) which proposes a minimax procedure that yields $L_2$ optimal transport maps.

## 3 EXPERIMENTS

**Setup** We follow the procedure in Algorithm 1. We start with drawing $(X_0, X_1) \sim \pi_0 \times \pi_1$ and use it to get the first rectified flow $\boldsymbol{Z}^1$ by minimizing (1). The second rectified flow $\boldsymbol{Z}^2$ is obtained by the same procedure except with the data replaced by the draws from $(Z_0^1, Z_1^1)$, generated by simulating the first rectified flow $\boldsymbol{Z}^1$. This process is repeated for $k$ times to get the $k$-*rectified flow* $\boldsymbol{Z}^k$. Finally, we can further distill the $k$-*rectified flow* $\boldsymbol{Z}^k$ into a one step model $z_1 = z_0 + v(z_0, 0)$ by fitting it on draws from $(Z_0^k, Z_1^k)$. By default, the ODEs are simulated using the vanilla Euler method with constant step size $1/N$ for $N$ steps, that is, $\hat{Z}_{t+1/N} = \hat{Z}_t + v(\hat{Z}_t, t)/N$ for $t \in \{0, \dots, N\}/N$. We use the Runge-Kutta

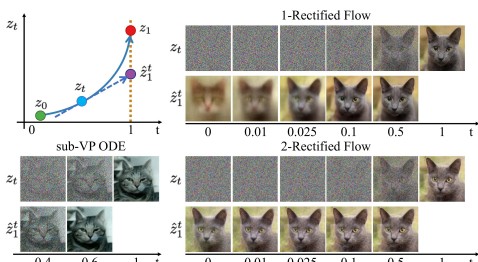

Figure 4: Sample trajectories $z_t$ of different flows on the AFHQ Cat dataset, and the extrapolation $\hat{z}_1^t = z_t + (1 - t)v(z_t, t)$ from different $z_t$. The same random seed is adopted for all methods. The $\hat{z}_1^t$ of 2-rectified flow is almost independent with $t$, indicating that its trajectory is almost straight.

method of order 5(4) from Scipy (Virtanen et al., 2020), denoted as RK45, which adaptively decide the step size and number of steps $N$ based on user-specified relative and absolute tolerances. In our experiments, we stick to the same parameters as Song et al. (2020b). More details can be found in Appendix E.

### 3.1 UNCONDITIONED IMAGE GENERATION

**Experiment settings** For generative modeling, we set $\pi_0$ to be the standard Gaussian distribution and $\pi_1$ the data distribution. Our implementation of rectified flow is modified upon the open-source code of (Song et al., 2020b). We adopt the U-Net architecture of DDPM++ (Song et al., 2020b) for representing the drift $v^X$, and report in Table 1 (a) and Figure 5 the results of our method and the (sub)-VP ODE from Song et al. (2020b) using the same architecture. Other recent results using different network architectures are shown in Table 1 (b) for reference.

**Results**  • *Results of fully solved ODEs.* As shown in Table 1 (a), the 1-rectified flow trained on the DDPM++ architecture, solved with RK45, yields the lowest FID (2.58) and highest recall (0.57) among all the ODE-based methods. In particular, the recall of 0.57 yields a substantial improvement over existing ODE and GAN methods.

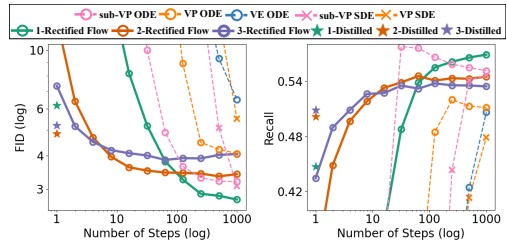 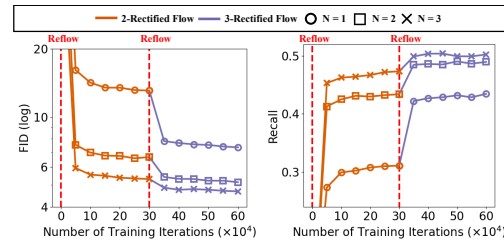

(a) FID and Recall vs. Number of Euler discretization steps $N$  (b) FID and Recall vs. Training Iterations

Figure 5: (a) Results of rectified flows and (sub-)VP ODE on CIFAR10 with different number $N$ of Euler discretization steps. (b) The FID and recall during different reflow and training steps. In (a), $k$-*Distilled* refers to the one-step model distilled from $k$-*Rectified Flow* for $k = 1, 2, 3$.

• *Results on few and single step generation.* As shown in Figure 5, the reflow procedure substantially improves both FID and recall in the small step regime (e.g., $N \lesssim 80$), even though it worsens the results in the large step regime due to the accumulation of error on estimating $v^x$. Figure 5 (b) show that each reflow leads to a noticeable improvement in FID and recall. For one-step generation ($N = 1$), the results are further boosted by distillation (see the stars in Figure 5 (a)). Overall, the distilled $k$-Rectified Flow with $k = 1, 2, 3$ yield one-step generative models beating all previous ODEs with distillation; they also beat the reported results of one-step models with similar U-net type architectures trained using GANs (see the *GAN with U-Net* in Table 1 (b)). In particular, the distilled 2-rectified flow achieves an FID of $4.85$, beating the best known one-step generative model with U-net architecture, $8.91$ (TDPM, Table 1 (b)). The recalls of both 2-rectified flow ($0.50$) and 3-rectified flow ($0.51$) outperform the best known results of GANs ($0.49$ from StyleGAN2+ADA) showing an advantage in diversity.

• *Reflow straightens the flow.* Figure 6 shows the reflow procedure decreases improves the straightness of the flow on CIFAR10. In Figure 4 visualizes the trajectories of 1-rectified flow and 2-rectified flow on the AFHQ cat dataset: at each point $z_t$, we extrapolate the terminal value at $t = 1$ by $\hat{z}_1^t = z_t + (1 - t)v(z_t, t)$; if the trajectory of ODE follows a straight line, $\hat{z}_1^t$ should not change as we vary $t$ when following the same path. We observe that $\hat{z}_1^t$ is almost independent with $t$ for 2-rectified flow, showing the path is almost straight. Moreover, even though 1-rectified flow is not straight with $\hat{z}_1^t$ over

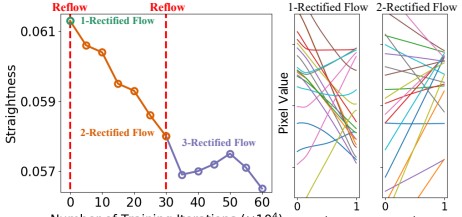

Figure 6: The straightening effect on CIFAR10. Left: the straightness measure *w.r.t.* reflow steps and iterations. Right: trajectories of randomly sampled pixels following 1- and 2-rectified flow.

time, it still yields recognizable and clear images very early ($t \approx 0.1$). In comparison, it is need $t \approx 0.6$ to get a clear image from the extrapolation of sub-VP ODE.

**High-resolution image generation** Figure 7 shows the result of 1-rectified flow on image generation on high-resolution ($256 \times 256$) datasets, including LSUN Bedroom (Yu et al., 2015), LSUN Church (Yu et al., 2015), CelebA HQ (Karras et al., 2018) to AFHQ Cat (Choi et al., 2020). Ours can generate high quality results across the different datasets. Figure 1 & 4 show that 1-(2-) rectified flow yields good results with one or few Euler steps.

## 3.2 Image-to-Image Translation

Assume we are given two sets of images of different styles (a.k.a. domains), whose distributions are denoted by $\pi_0, \pi_1$, respectively. We are interested in transferring the style (or other key characteristics) of the images in one domain to the other domain, in the absence of paired examples. A classical approach to achieving this is cycle-consistent adversarial networks (a.k.a. CycleGAN) (Zhu et al., 2017; Isola et al., 2017), which jointly learns a forward and backward mapping $F, G$ by minimizing the sum of adversarial losses on the two domains, regularized by a cycle consistency loss to enforce $F(G(x)) \approx x$ for all image $x$. By constructing the rectified flow of $\pi_0$ and $\pi_1$, we obtain a simple approach to image translation that requires no adversarial optimization and cycle-consistency regularization: training the rectified flow requires a simple optimization procedure and the cycle consistency is automatically in flow models satisfied due to reversibility of ODEs.

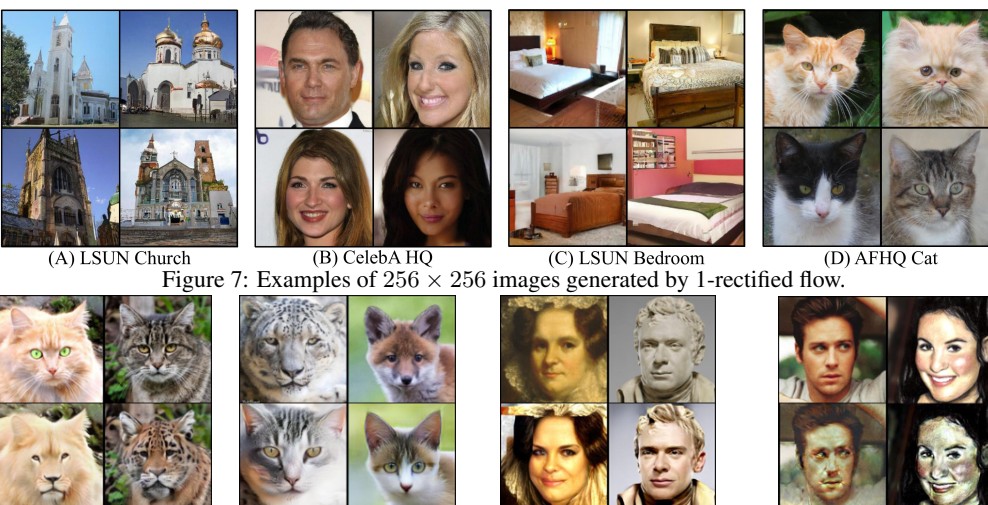

(A) LSUN Church      (B) CelebA HQ      (C) LSUN Bedroom      (D) AFHQ Cat

Figure 7: Examples of $256 \times 256$ images generated by 1-rectified flow.

(A) Cat $\to$ Wild Animals    (B) Wild Animals $\to$ Cat    (C) MetFace $\to$ CelebA Face    (D) CelebA Face $\to$ MetFace

Figure 8: Samples of 1-rectified flow simulated with $N = 100$ Euler steps between different domains.

As the main goal here is to obtain good visual results, we are not interested in faithfully transferring $X_0 \sim \pi_0$ to an $X_1$ that exactly follows $\pi_1$. Rather, we are interested in transferring the image styles while preserving the identity of the main object in the image. For example, when transferring a human face image to a cat face, we are interested in getting a unrealistic face of human-cat hybrid that still "looks like" the original human face. To achieve this, let $h(x)$ be a feature mapping of image $x$ representing the styles that we are interested in transferring. Let $X_t = tX_1 + (1-t)X_0$. $H_t = h(X_t)$ follows an ODE of $\mathrm{d}H_t = \nabla h(X_t)^\top (X_1 - X_0)\mathrm{d}t$. Hence, to ensure that the style is transferred correctly, we propose to learn $v$ such that $H_t = h(Z_t)$ with $\mathrm{d}Z_t = v(Z_t, t)\mathrm{d}t$ approximates $H_t$ as much as possible. Because $\mathrm{d}H'_t = \nabla h(Z_t)^\top v(Z_t, t)\mathrm{d}t$, we minimize the loss:

$$\min_v \int_0^1 \mathbb{E}\left[\left\|\nabla h(X_t)^\top (X_1 - X_0 - v(X_t, t))\right\|_2^2\right]\mathrm{d}t, \qquad X_t = tX_1 + (1-t)X_0. \qquad (4)$$

In practice, we set $h(x)$ to be latent representation of a classifier trained to distinguish the images from the two domains $\pi_0, \pi_1$, fine-tuned from a pre-trained ImageNet (Tan & Le, 2019) model. Intuitively, $\nabla_x h(x)$ serves as a saliency score and re-weights coordinates so that the loss in (4) focuses on penalizing the error that causes significant changes on $h$.

Initialization   1-Rectified Flow   2-Rectified Flow   1-Rectified Flow   2-Rectified Flow   Initialization   1-Rectified Flow   2-Rectified Flow   1-Rectified Flow   2-Rectified Flow
   $N = 100$    $N = 100$    $N = 1$    $N = 1$      $N = 100$    $N = 100$    $N = 1$    $N = 1$

Figure 9: Samples of results of 1- and 2-rectified flow simulated with $N = 1$ and $N = 100$ Euler steps.

**Experiment settings**      We set the domains $\pi_0, \pi_1$ to be pairs of the AFHQ (Choi et al., 2020), MetFace (Karras et al., 2020) and CelebA-HQ (Karras et al., 2018) dataset. The results are shown by initializing the trained flows from the test data. The training and network configurations follow Section 3.1. See Appendix E for details.

**Results**      Figure 1, 8, 9 show examples of results of 1- and 2-rectified flow simulated with Euler method with different number of steps $N$. We can see that rectified flows can successfully transfer the styles and generate high quality images. Moreover, 2-rectified flow returns good results with a single Euler step. See more examples in Appendix E.

## 4   CONCLUSIONS

Rectified flow provides a simple and clean framework for learning transport mappings from data: (1) it can be applied to both generative and transfer modeling; (2) it is able to learn fast ODEs (even one-step) by learning straight flows; (3) it provides a new way for understanding the diffusion models and their ODE variants; (4) it is purely ODE-based, avoiding SDE models both conceptually and algorithmically; (5) the theoretical and algorithmic insights to optimal transport are of independent interest; (6) the idea of causalizing interpolation processes provides a general framework related to transport mapping problems and is amenable to rigorous theoretical analysis.

## ACKNOWLEDGEMENTS

This research is supported by NSF CAREER1846421, SenSE2037267, EAGER-2041327, Office of Navy Research, and NSF AI Institute for Foundations of Machine Learning (IFML).

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

**Roadmap**  The appendix is structured as follows:
- Appendix A discusses related works on generative models, neural ODE, diffusion models, and optimal transport.
- Appendix B discusses the form and estimation of velocity field $v^X$ of rectified flow.
- Appendix C introduces a nonlinear extension of our method, which include probability flow ODEs (Song et al., 2020b) and DDIM (Song et al., 2020a) as special cases.
- Appendix D presents the full course theoretical analysis of rectified flow.
- Appendix E gives the details of the experiments and additional experiment results.

# A  RELATED WORKS AND DISCUSSION

**Learning one-step models**  GANs (Goodfellow et al., 2014; Arjovsky et al., 2017; Liu et al., 2021), VAEs (Kingma & Welling, 2013), and (discrete-time) normalizing flows (Rezende & Mohamed, 2015; Dinh et al., 2014; 2016) have been three classical approaches for learning deep generative models. GANs have been most successful in terms of generation qualities (for images in particular), but suffer from the notorious training instability and mode collapse issues due to use of minimax updates. VAEs and normalizing flows are both trained based on the principle of maximum likelihood estimation (MLE) and need to introduce constraints on the model architecture and/or special approximation techniques to ensure tractable likelihood computation: VAEs typically use a conditional Gaussian distribution in addition to the variational approximation of the likelihood; normalizing flows require to use specially designed invertible architectures and need to copy with calculating expensive Jacobian matrices.

The reflow+distillation approach in this work provides another promising approach to training one-step models, avoiding the minimax issues of GANs and the intractability issues of the likelihood-based methods.

**Learning ODEs: MLE and PF-ODEs**  There are two major approaches for learning neural ODEs: the PF-ODEs/DDIM approach discussed in Section C, and the more classical MLE based approach of Chen et al. (2018).

• *The MLE approach.* In Chen et al. (2018), neural ODEs are trained for learning generative models by maximizing the likelihood of the distribution of the ODE outcome $Z_1$ at time $t = 1$ under the data distribution $\pi_1$. Specifically, with observations from $\pi_1$, it estimates a neural drift $v$ of an ODE $\mathrm{d}Z_t = v(Z_t, t)\mathrm{d}t$ by

$$\min_v \mathbb{D}(\pi_1; \ \rho^{v, \pi_0}), \tag{5}$$

where $\mathbb{D}(\cdot; \ \cdot)$ denotes KL divergence (or other discrepancy measures), and $\rho^{v, \pi_0}$ is the density of $Z_1$ following $\mathrm{d}Z_t = v(Z_t, t)\mathrm{d}t$ from $Z_0 \sim \pi_0$; the density of $\pi_0$ should be known and tractable to calculate.

By using an instantaneous change of variables formula, it was observed in Chen et al. (2018) that the likelihood of neural ODEs are easier to compute than the discrete-time normalizing flow without constraints on the model structures. However, this MLE approach is still computationally expensive for large scale models as it requires repeated simulation of the ODE during each training step. In addition, as the optimization procedure of MLE requires to backpropagate through time, it can easily suffer the gradient vanishing/exploding problem unless proper regularization is added.

Another fundamental problem is that the MLE (5) of neural ODEs is theoretically under-specified, because MLE only concerns matching the law of the final outcome $Z_1$ with the data distribution $\pi_1$, and there are infinitely many ODEs to achieve the same output law of $Z_1$ while traveling through different paths. A number of works have been proposed to remedy this by adding regularization terms, such as these based on transport costs, to favor shorter paths; see (Nichol & Dhariwal, 2021; Onken et al., 2021). Without a regularization term, the ODE learned by MLE would be implicitly determined by the initialization and other hyper-parameters of the optimizer used to solve (5).

• *Probability Flow ODEs.* The method of PF-ODEs (Song et al., 2020b) and DDIM (Song et al., 2020a) provides a different approach to learning ODEs that avoids the main disadvantages of the MLE approach, including the expensive likelihood calculation, training-time simulation of the ODE models, and the need of backpropagation through time. However, because PF-ODEs and DDIM

were derived as the side product of learning the mathematically more involved diffusion/SDE models, their theories and algorithm forms were made unnecessarily restrictive and complicated. The nonlinear rectified flow framework shows that the learning of ODEs can be approached directly in a very simple way, allowing us to identify the canonical case of linear rectified flow and open the door of further improvements with flexible and decoupled choices of the interpolation curves $X_t$ and initial distributions $\pi_0$.

Viewed through the general non-linear rectified flow framework, the computational and theoretical drawbacks of MLE can be avoided because we can simply pre-determines the "roads" that the ODEs should travel through by specifying the interpolation curve $X_t$, rather than leaving it for the algorithm to figure out implicitly. It is theoretically valid to pre-specify any interpolation $X_t$ because the neural ODE is highly over-parameterized as a generative model: when $v$ is a universal approximator and $\pi_0$ is absolutely continuous, the distribution of $Z_1$ can approximate any distribution given any fixed interpolation curve $X_t$. The idea of rectified flow is to the simplest geodesic paths for $X_t$.

**Learning SDEs with denoising diffusion**   Although the scope of this work is limited to learning ODEs, the score-based generative models (Song & Ermon, 2019; 2020; Song et al., 2020b; 2021) and denoising diffusion probability models (DDPM) (Ho et al., 2020) are of high relevance as the basis of PF-ODEs and DDIM. The diffusion/SDE models trained with these methods have been found outperforming GANs in image synthesis in both quality and diversity (Dhariwal & Nichol, 2021). Notably, thanks to the stable and scalable optimization-based training procedure, the diffusion models have successfully used in huge text-to-image generation models with astonishing results (e.g., Nichol et al., 2021; Ramesh et al., 2022; Saharia et al., 2022). It has been quickly popularized in other domains, such as video (e.g., Ho et al., 2022; Yang et al., 2022; Harvey et al., 2022), music (Mittal et al., 2021), audio (e.g., Kong et al., 2020; Lee & Han, 2021; Popov et al., 2021), and text (Li et al., 2022; Wang et al., 2022), and more tasks such as image editing (Zhao et al., 2022; Meng et al., 2021; Zhang et al., 2022a). A growing literature has been developed for improving the inference speed of denoising diffusion models, an example of which is the PF-ODEs/DDIM approach which gains speedup by turning SDEs into ODEs. We provide below some examples of recent works, which is by no mean exhaustive.

● *Improved training and inference.* A line of works focus on improving the inference and sampling procedure of denoising diffusion models. For example, Nichol & Dhariwal (2021) presents a few simple modifications of DDPM to improve the likelihood, sampling speed, and generation quality. Karras et al. (2022) systematic exams the design space of diffusion generative models with empirical studies and identifies a number of training and inference recipes for better generative quality with fewer sampling steps. Zhang & Chen (2022) proposes a diffusion exponential integrator sampler for fast sampling of diffusion models. Lu et al. (2022) provides a customized high order solver for PF-ODEs. Bao et al. (2022) provides an analytic estimate of the optimal diffusion coefficient.

● *Combination with other methods.* Another direction is to speed up diffusion models by combining them with GANs and other generative models. DDPM Distillation (Luhman & Luhman, 2021) accelerates the inference speed by distilling the trajectories of a diffusion model into a series of conditional GANs. The truncated diffusion probabilistic model (TDPM) of (Zheng et al., 2022) trains a GAN model as $\pi_0$ so that the diffusion process can be truncated to improve the speed; the similar idea was explored in Lyu et al. (2022); Franzese et al. (2022), and (Franzese et al., 2022) provides an analysis on the optimal truncation time. (Sinha et al., 2021; Wehenkel & Louppe, 2021; Vahdat et al., 2021) learns a denoising diffusion model in the latent spaces and combines it with variational auto-encoders. These methods can be potentially applied to rectified flow to gain similar speedups for learning neural ODEs.

● *Unpaired Image-to-Image translation.* The standard denoising diffusion and PF-ODEs methods focus on the generative task of transferring a Gaussian noise ($\pi_0$) to the data ($\pi_1$). A number of works have been proposed to adapt it to transferring data between arbitrary pairs of source-target domains. For example, SDEdit Meng et al. (2021) synthesizes realistic images guided by an input image by first adding noising to the input and then denoising the resulting image through a pre-trained SDE model. Choi et al. (2021) proposes a method to guide the generative process of DDPM to generate realistic images based on a given reference image. Su et al. (2022) leverages two two PF-ODEs for image translation, one translating source images to a latent variable, and the other constructing the target images from the latent variable. Zhao et al. (2022) proposes an energy-guided approach

that employs an energy function pre-trained on the source and target domains to guide the inference process of a pretrained SDE for better image translation. In comparison, our framework shows that domain transfer can be achieved by essentially the same algorithm as generative modeling, by simply setting $\pi_0$ to be the source domain.

• *Diffusion bridges.* Some recent works Peluchetti (2021); Liu et al. (2022) show that the design space of denoising diffusion models can be made highly flexible with the assistant of diffusion bridge processes that are pinned to a fixed data point at the end time. This reduces the design of denoising diffusion methods to constructing a proper bridge processes. The bridges in Song et al. (2020b) are constructed by a time-reversal technique, which can be equivalently achieved by Doob's $h$-transform as shown in Peluchetti (2021); Liu et al. (2022), and more general construction techniques are discussed in Liu et al. (2022); Wu et al. (2022). Despite the significantly extended design spaces, an unanswered question is what type of diffusion bridge processes should be preferred. This question is made challenging because the presence of diffusion noise and the need of advanced stochastic calculus tools make it hard to intuit how the methods work. By removing the diffusion noise, our work makes it clear that straight paths should be preferred. We expect that the idea can be extended to provide guidance on designing optimal bridge processes for learning SDEs.

• *Schrodinger bridges.* Another body of works (Wang et al., 2021; De Bortoli et al., 2021; Chen et al., 2021; Vargas et al., 2021) leverages Schrodinger bridges (SB) as an alternative approach to learning diffusion generative models. These approaches are attractive theoretically, but casts significant computational challenges for solving the Schrodinger bridge problem.

**Re-thinking the role of diffusion noise** The introduction of diffusion noise was consider essential due to the key role it plays in the derivations of the successful methods (Song et al., 2020b; Ho et al., 2020). However, as rectified flow can achieve better or comparable results with a ODE-only framework, the role of diffusion mechanisms should be re-examined and clearly decoupled from the other merits of denoising diffusion models. The success of the denoising diffusion models may be mainly attributed to the simple and stable optimization-based training procedure that allows us to avoid the instability issues and the need of case-by-case tuning of GANs, rather than the presence of diffusion noises.

Because our work shows that there is no need to invoke SDE tools if the goal is to learn ODEs, the remaining question is whether we should learn an ODE or an SDE for a given problem. As already argued by a number of works (Song et al., 2020b;a; Karras et al., 2022), ODEs should be preferred over SDEs in general. Below is a detailed comparison between ODEs and SDEs.

• *Conceptual simplicity and numerical speed.* SDEs are more mathematically involved and are more difficult to understand. Numerical simulation of ODEs are simpler and faster than SDEs.

• *Time reversibility.* It is equally easy to solve the ODEs forwardly and backwardly. In comparison, the time reversal of SDEs (e.g., Anderson, 1982; Haussmann & Pardoux, 1986; Föllmer, 1985) is more involved theoretically and may not be computationally tractable.

• *Latent spaces.* The couplings $(Z_0, Z_1)$ of ODEs are deterministic and yield low transport cost in the case of rectified flows, hence providing a good latent space for representing and manipulating outputs. Introducing diffusion noises make $(Z_0, Z_1)$ more stochastic and hence less useful. In fact, the $(Z_0, Z_1)$ given by DDPM Ho et al. (2020) and the SDEs of Song et al. (2020b) and hence useless for latent presentation.

• *Training difficulty.* There is no reason to believe that training an ODE is harder, if not easier, than training an SDE sharing the same marginal laws: the training loss of both cases would share the distributions of covariant and differ only on the targets. In the setting of (Song et al., 2020b), the loss functions are equivalent up to a linear reparameterization.

• *Expressive power.* As every SDE can be converted into an ODE that has the same marginal distribution using the techniques in Song et al. (2020a;b) (see also Villani (2009)), ODEs are as powerful as SDEs for representing marginal distributions, which is what needed for the transport mapping problems considered in this work. On the other hand, SDEs may be preferred if we need to capture richer time-correlation structures.

• *Manifold data.* When equipped with neural network drifts, the outputs of ODEs tend to fall into a smooth low dimensional manifold, a key inductive for structured data in AI such as images and text.

In comparison, when using SDEs to model manifold data, one has to carefully anneal the diffusion noise to obtain smooth outcomes, which causes slow computation and a burden of hyperparameter tuning. SDEs might be more useful in for modeling highly noisy data in areas like finance and economics, and in areas that involve diffusion processes physically, such as molecule simulation.

**Optimal vs. straight transport**   Optimal transport has been extensively explored in machine learning as a powerful way to compare and transfer between probability measures. For the transport mapping problem considered in this work, a natural approach is to finding the optimal coupling $(Z_0, Z_1)$ that minimizes a transport cost $\mathbb{E}[c(Z_1 - Z_0)]$ for a given $c$. The most common choice of $c$ is the quadratic cost $c(\cdot) = \|\cdot\|^2$.

However, finding the optimal couplings, especially for high dimensional continuous measures, is highly challenging computationally and is the subject of active research; see for example (Seguy et al., 2017; Korotin et al., 2021; 2022; Makkuva et al., 2020; Rout et al., 2021; Daniels et al., 2021). In addition, although the optimal couplings are known to have nice smoothness and other regularity properties, it is not necessary to accurately find the optimal coupling because the transport cost do not exactly align with the learning performance of individual problems; see e.g., Korotin et al. (2021).

In comparison, our reflow procedure finds a straight coupling, which is not optimal w.r.t. a given $c$ (see Section D.4). From the perspective of fast inference, all straight couplings are equally good because they all yield straight rectified flows and hence can be simulated with one Euler step.

---

**Algorithm 2** `Train(Data)`

```
# Input:   Data={x0, x1}
# Output:  Model v(x,t) for the rectified flow
initialize Model
for x0, x1 in Data:  # x0, x1: samples from π₀, π₁
    Optimizer.zero_grad()
    t = torch.rand(batchsize)  # Randomly sample t ∈ [0,1]
    Loss = ( Model(t*x1+(1-t)*x0, t) - (x1-x0) ).pow(2).mean()
    Loss.backward()
    Optimizer.step()
return Model
```

---

**Algorithm 3** `Sample(Model, Data)`

```
# Input:   Model v(x,t) of the rectified flow
# Output:  draws of the rectified coupling (Z₀,Z₁)
coupling = []
for x0, _ in Data:  # x0: samples from π₀ (batchsize×dim)
    x1 = model.ODE_solver(x0)
    coupling.append((x0, x1))
return coupling
```

---

**Algorithm 4** `Reflow(Data)`

```
# Input:   Data={x0, x1}
# Output:  draws of the K-th rectified coupling
Coupling = Data
for k = 1,...,K:
    Model = Train(Coupling)
    Coupling = sample(Model, Data)
return Coupling
```

---

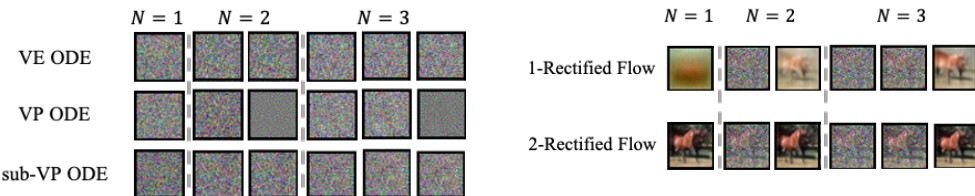

Figure 10: Few-step generation with different ODEs. Compared with VE,VP,sub-VP ODE, 1-rectified flow can generate blurry images using only 1,2,3 steps. After one time of rectification, 2-rectified flow can generate clear images with 1,2,3 steps.

## B ON THE VELOCITY FIELD $v^X$ OF RECTIFIED FLOW

**On the velocity field $v^X$**   If $X_0$ yields a conditional density function $\rho(x_0 \mid x_1)$ when conditioned on $X_1 = x_1$, then the optimal velocity field $v^X(z, t) = \mathbb{E}[X_1 - X_0 | X_t = z]$ can be represented by

$$v^X(z, t) = \mathbb{E}\left[\frac{X_1 - z}{1 - t}\eta_t(X_1, z)\right], \quad \eta_t(X_1, z) = \rho\left(\frac{z - tX_1}{1 - t} \,\Big|\, X_1\right) \Big/ \mathbb{E}\left[\rho\left(\frac{z - tX_1}{1 - t} \,\Big|\, X_1\right)\right], \tag{6}$$

where the expectation $\mathbb{E}[\cdot]$ is taken w.r.t. $X_1 \sim \pi_1$. This can be seen by noting that $X_0 = \frac{z - tX_1}{1-t}$ and $X_1 - X_0 = \frac{X_1 - z}{1-t}$, when conditioned on $X_t = z$. Hence, if $\rho$ is positive and continuous everywhere, then $v^X$ is well defined and continuous on $\mathbb{R}^d \times [0, 1)$. Further, if $\log \eta_t$ is continuously differentiable w.r.t. $z$, we can show that

$$\nabla_z v^X(z, t) = \frac{1}{1 - t}\mathbb{E}\left[\left((X_1 - z)\nabla_z \log \eta_t(X_1, z) - 1\right)\eta_t(X_1, z)\right].$$

Note that $\mathrm{d}Z_t = v^X(Z_t, t)\mathrm{d}t$ is guaranteed to have a unique solution if $v^X$ is uniformly Lipschitz continuous on $[0, a]$ for any $a < 1$.

If $X_0|X_1 = x_1$ does not yield a conditional density function, $v^X(z, t)$ may be undefined or discontinuous, making the ODE $\mathrm{d}Z_t = v^X(Z_t, t)\mathrm{d}t$ ill-behaved. A simple fix is to add $X_0$ with a Gaussian noise $\xi \sim \mathcal{N}(0, \sigma^2 I)$ independent of $(X_0, X_1)$ to yield a smoothed variable $\tilde{X}_0 = X_0 + \xi$, and transfer $\tilde{X}_0$ to $X_1$ using rectified flow. This would effectively give a randomized mapping of form $T(X_0 + \xi)$ transporting $\pi_0$ to $\pi_1$.

**Smooth function approximation**   Following (6), we can *exactly* calculate $v^X$ if the conditional density function $\rho(\cdot|x_1)$ exists and is known, and $\pi_1$ is the empirical measure of a finite number of points (whose expectation can be evaluated exactly). In this case, running the rectified flow forwardly would precisely recover the points in $\pi_1$. This, however, is not practically useful in most cases as it completely overfits the data. Hence, it is both necessary and beneficial to fit $v^X$ with a smooth function approximator such as neural network or non-parametric models, to obtain smoothed distributions with novel samples that are practically useful.

Deep neural networks are no doubt the best function approximators for large scale problems. For low dimensional problems, the following simple Nadaraya–Watson style non-parametric estimator of $v^X$ can yield a good approximation to the exact rectified flow without knowing the conditional density $\rho$:

$$v^{X,h}(z, t) = \mathbb{E}\left[\frac{X_1 - z}{1 - t}\omega_h(X_t, z)\right], \tag{7}$$

where $\omega_h(X_t, z) = \frac{\kappa_h(X_t, z)}{\mathbb{E}[\kappa_h(X_t, z)]}$, and $\kappa_h(x, z)$ is a smoothing kernel with a bandwith parameter $h > 0$ that measures the similarity between $z$ and $x$. Taking the Gaussian RBF kernel $\kappa_h(x, z) = \exp(-\|x - z\|^2 /2h^2)$, then when $h \to 0^+$, it can be shown that $v^{X,h}(z, t)$ converges to $v^X(z, t) = \mathbb{E}\left[\frac{X_1 - z}{1 - t} \mid X_t = z\right]$ on points $z$ that can be attained by $X_t$ (i.e., the conditional expectation $\mathbb{E}[\cdot \mid X_t = z]$ exists. ). On points $z$ that $X_t$ can not attain, $v^{X,h}(z, t)$ extrapolates the value

by finding the $X_t$ that is close to $z$. In practice, we replace the expectations in (7) with empirical averaging. We find that $v^{X,h}$ performs well in practice because it is a mixture of linear functions that always point to a point in the support of $\pi_1$.

## C  A Nonlinear Extension of Rectified Flow

We present a nonlinear extension of rectified flow in which the linear interpolation $X_t$ is replaced by any time-differentiable curve connecting $X_0$ and $X_1$. Such generalized rectified flows can still transport $\pi_0$ to $\pi_1$ (Theorem D.3), but no longer guarantee to decrease convex transport costs, or have the straightening effect. Importantly, the method of probability flows (Song et al., 2020b) and DDIM (Song et al., 2020a) can be viewed (approximately) as special cases of this framework, allows us to clarify the connection with and the advantages over these methods.

Let $\boldsymbol{X} = \{X_t \colon t \in [0,1]\}$ be any time-differentiable random process that connects $X_0$ and $X_1$. Let $\dot{X}_t$ be the time derivative of $X_t$. More specifically, as what is standard in stochastic process, the stochastic process $X_t$ can be viewed as a measurable function $X_t = X(t, \omega)$, where $\omega$ can be viewed as a some "random seed". The time-derivative is $\dot{X}_t = \partial_t X(t, \omega)$.

The (nonlinear) rectified flow induced from $\boldsymbol{X}$ is defined as

$$\mathrm{d}Z_t = v^{\boldsymbol{X}}(Z_t, t)\mathrm{d}t, \quad \text{with} \quad Z_0 = X_0, \quad \text{and} \quad v^{\boldsymbol{X}}(z, t) = \mathbb{E}\left[\dot{X}_t \mid X_t = t\right].$$

We can estimate $v^{\boldsymbol{X}}$ by solving

$$\min_v \int_0^1 \mathbb{E}\left[w_t \left\| v(X_t,\ t) - \dot{X}_t \right\|^2\right] \mathrm{d}t, \tag{8}$$

where $w_t \colon (0,1) \to (0, +\infty)$ is a positive weighting sequence ($w_t = 1$ by default). When using the linear interpolation $X_t = tX_1 + (1-t)X_0$, we have $\dot{X}_t = X_1 - X_0$ and (8) with $w_t = 1$ reduces to (1). As we show in Theorem D.3, the flow $\boldsymbol{Z}$ given by this method still preserves the marginal laws of $\boldsymbol{X}$, that is, $\mathrm{Law}(Z_t) = \mathrm{Law}(X_t), \forall t \in [0,1]$, and hence $(Z_0, Z_1)$ remains to be a coupling of $\pi_0, \pi_1$. However, if $\boldsymbol{X}$ is not straight, $(Z_0, Z_1)$ no longer guarantees to decrease the convex transport costs over $(X_0, X_1)$. More importantly, the reflow procedure no longer straightens the paths of $Z_t$.

A simple class of interpolation processes is $X_t = \alpha_t X_1 + \beta_t X_0$ where $\alpha_t$ and $\beta_t$ are two differentiable sequences that satisfy $\alpha_1 = \beta_0 = 1$ and $\alpha_0 = \beta_1 = 0$ to ensure that the process equals $X_0, X_1$ at the starting and end points. In this case, we have $\dot{X}_t = \dot{\alpha}_t X_1 + \dot{\beta}_t X_0$ in (8) where $\dot{\alpha}_t$ and $\dot{\beta}_t$ are the time derivatives of $\alpha_t$ and $\beta_t$. The shape of the curve is controlled by the relation of $\alpha_t$ and $\beta_t$. If we take $\beta_t = 1 - \alpha_t$ for all $t$, then $X_t$ have straight paths but does not travel at a constant speed; it can be viewed as a time-changed variant of the canonical case $X_t = tX_1 + (1-t)X_0$ when $t$ is reparameterized to $\alpha_t$. When $\beta_t \neq 1 - \alpha_t$, the paths of $X_t$ are not straight lines except some special cases (e.g., $\dot{\alpha}_t X_1 = 0$ or $\dot{\beta}_t X_0 = 0$, or $X_1 = aX_1$ for some $a \in \mathbb{R}$).

### C.1  Probability Flow ODEs and DDIM

The probability flow ODEs (PF-ODEs) Song et al. (2020b) and denoising diffusion implicit models (DDIM) Song et al. (2020a) are methods for learning ODE-based generative models of $\pi_1$ from a spherical Gaussian initial distribution $\pi_0$, derived by converting a SDE learned by denoising diffusion methods to an ODE with equivalent marginal laws. In Song et al. (2020b), three types of PF-ODEs are derived from three types of SDEs learned as score-based generative models, including variance-exploding (VE) SDE, variance-preserving (VP) SDE, and sub-VP SDE, which we denote by VE ODE, VP ODE, and sub-VP ODE, respectively. VP ODE is equivalent to the continuous time limit of DDIM, which is derived from the denoising diffusion probability model (DDPM) Ho et al. (2020). As the derivations of PF-ODEs and DDIM require advanced tools in stochastic calculus, we limit our discussion on the final algorithmic procedures suggested in Song et al. (2020b); Ho et al. (2020). The readers are referred to Song et al. (2020b;a) for the details.

**Proposition C.1.** *All variants of PF-ODEs can be viewed as instances of* (8) *when using* $X_t = \alpha_t X_1 + \beta_t \xi$ *for some* $\alpha_t, \beta_t$ *with* $\alpha_1 = 1, \beta_1 = 0$, *where* $\xi \sim \mathcal{N}(0, I)$ *is a standard Gaussian random variable.*

Here we need to use introduce $\xi$ to replace $X_0$ because the choices of $\alpha_t$ and $\beta_t$ suggested in Song et al. (2020b;a); Ho et al. (2020) do not satisfy the boundary condition of $\alpha_0 = 0$ and $\beta_0 = 1$ at $t = 0$, and hence $X_0 \neq \xi$. Instead, in these methods, the initial distribution $X_0 \sim \pi_0$ is implicitly defined as $X_0 = \alpha_0 X_1 + \beta_0 \xi$, which is approximated by $X_0 \approx \beta_0 \xi$ by making $\alpha_0 X_1 \ll \beta_0 \xi$. Hence, $\pi_0$ is set to be $\mathcal{N}(0, \beta_0^2 I)$ in these methods. Viewed through our framework, there is no reason to restrict $\xi$ to be $\mathcal{N}(0, \beta_0^2 I)$, or not set $\alpha_0 = 0, \beta_0 = 1$ to avoid the approximation.

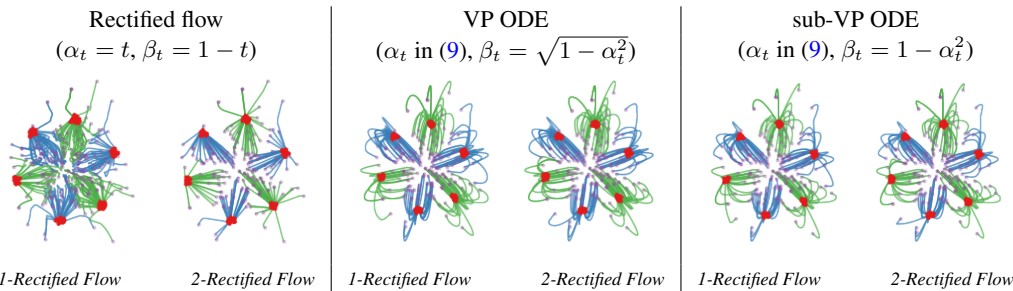

Figure 11: Comparing rectified flow with VP ODE and sub-VP ODE when $\pi_0 = \mathcal{N}(0, I)$ (purple dots) and $\pi_1$ is a low variance Gaussian mixture shown as the red dots. The linear rectified flow yields nearly straight trajectories with one step of reflow. But the trajectories of VP ODE and sub-VP ODE are curved and can not be straightened by reflowing.

| Time-Discretization Steps $N$ | Rectified Flow $\alpha_t = t, \beta_t = 1 - t$ | VP ODE $\alpha_t$ in (9), $\beta_t = \sqrt{1 - \alpha_t^2}$ | sub-VP ODE $\alpha_t$ in (9), $\beta_t = 1 - \alpha_t^2$ | VP ODE (const speed) $\alpha_t = t, \beta_t = \sqrt{1 - \alpha_t^2}$ |
|---|---|---|---|---|
| $N = 1$ | | | | |
| $N = 2$ | | | | |
| $N = 5$ | | | | |
| $N = 100$ | | | | |

Figure 12: Trajectories of different methods when varying the number of discretization steps $N$ (purple dots: $\pi_0$; red dots: $\pi_1$; orangle dots: intermediate steps; blue curves: flow trajectories). The rectified flow travels in straight lines and progresses uniformly in time; it generates the mean of $\pi_1$ when simulated with a single Euler step, and quickly covers the whole distribution $\pi_1$ with more steps (in this case $N = 2$ is sufficient). In comparison, VP ODE and sub-VP ODE travel in curves with non-uniform speed: they tend to be slow in the beginning and speed up in the later phase (much of the update happens when $t \gtrsim 0.5$). The non-uniform speed can be avoided by setting $\alpha_t = t$ (see the last column).

**VP ODE and sub-VP ODE**    The VP ODE and sub-VP ODE of Song et al. (2020b) use the following shared $\alpha_t$:

$$\text{(sub-)VP ODE:} \quad \alpha_t = \exp\left(-\frac{1}{4}a(1 - t)^2 - \frac{1}{2}b(1 - t)\right); \quad \text{default values: } a = 19.9, b = 0.1, \tag{9}$$

where the default values of $a, b$ are chosen to match the continuous time limit of the shared training procedure of DDIM and DDPM. The difference of VP ODE and sub-VP ODE is on the choice of $\beta_t$, given as follows:

$$\text{VP ODE:} \quad \beta_t = \sqrt{1 - \alpha_t^2}, \qquad\qquad \text{sub-VP ODE:} \quad \beta_t = 1 - \alpha_t^2. \tag{10}$$

As $\beta_0 \approx 1$ in both VP and sub-VP ODE, the $\pi_0$ in both cases are taken as $\mathcal{N}(0, I)$.

The choices of $\alpha_t, \beta_t$ above are the consequence of the SDE-based derivation in Song et al. (2020b). However, they are not well-motivated when we exam the path properties of the induced ODEs:

• *Non-straight paths:* Due to choices of $\beta_t$ in (10), the trajectories of VP ODE and sub-VP ODE are curved in general, and can not be straightened by the reflow procedure. We should choose $\beta_t = 1 - \alpha_t$ to induce straight paths.

• *Non-uniform speed:* The exponential form of $\alpha_t$ in (9) is a consequence of using Ornstein–Uhlenbeck processes in the derivation of SDE models (Song et al., 2020b; Ho et al., 2020). However, there is no clear advantage of using (9) for ODEs. As shown in Figure 12, the $\alpha_t$ and $\beta_t$ of VP and sub-VP ODE change slowly in the early phase ($t \lesssim 0.5$). As a result, the flow also moves slowly in beginning and hence most of the updates are concentrated in the later phase. Such non-uniform update speed, in addition to the non-straight paths, make VP ODE and sub-VP ODE perform sub-optimally when using large step sizes,

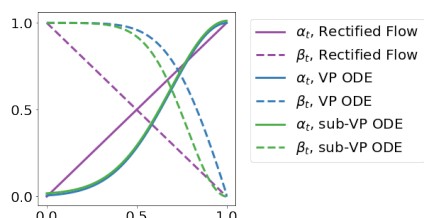

Figure 13: $t$ vs. $\alpha_t, \beta_t$ of different methods.

even for transport between simple spherical Gaussian distributions (see Figure 12). As we show in the last column of Figure 12, changing the exponential $\alpha_t$ to the linear function $\alpha_t = t$ in VP ODE allows us to get a uniform update speed while preserving the same continuous-time trajectories.

**VE ODE** The VE ODE of Song et al. (2020b) uses $\alpha_t = 1$ and $\beta_t = \sigma_{\min}\sqrt{r^{2(1-t)} - 1}$ where $\sigma_{\min} = 0.01$ by default $r$ is set such that $\sigma_{\max} := r\sigma_{\min}$ is as large as the maximum Euclidean distance between all pairs of training data points from $\pi_1$ (Technique 1 of Song & Ermon (2020)). Assume that $\sigma_{\max}^2$ is much larger than both $\sigma_{\min}^2$ and the variance of $X_1$, then $X_0 = X_1 + \beta_0 \xi \approx \sigma_{\max}\xi$, and we can set the initial distribution to be $\pi_0 \sim \mathcal{N}(0, \sigma_{\max}^2 I)$, which has much larger variance than $\pi_1$. Hence, VE ODE can not be applied to (and not shown in) the toys in Figure 11 and Figure 12. As the case of (sub-)VP ODE, the restriction on $\xi$ is in fact unnecessary and requirement that $\sigma_{\max}$ is unnatural viewed from our framework. On the other hand, the trajectories of $X_t$ in VE ODE are indeed straight lines, because the direction of $\dot{X}_t = \dot{\beta}_t \xi$ is always the same as $\xi$. However, the choice of $\beta_t$ causes a non-uniform speed issue similar to that of (sub-)VP ODE.

Following Song et al. (2020b); Ho et al. (2020), a line of works have been proposed to improve the choices of $\alpha_t, \beta_t$, but remain to be constrained by the basic design space from the SDE-to-ODE derivation; see for example Nichol & Dhariwal (2021); Karras et al. (2022); Zhang et al. (2022b).

To summarize, the simple nonlinear rectified flow framework in (8) both simplifies and extends the existing framework, and sheds a number of importance insights:

• Learning ODEs can be considered directly and independently without resorting to diffusion/SDE methods;

• The paths of the learned ODEs can be specified by any smooth interpolation curve $X_t$ of $X_0$ and $X_1$;

• The initial distribution $\pi_0$ can be chosen arbitrarily, independent with the choice of the interpolation $X_t$.

• The canonical linear interpolation $X_t = tX_1 + (1-t)X_0$ should be recommended as a default choice.

On the other hand, non-linear choices of $X_t$ can be useful when we want to incorporate certain non-Euclidan geometry structure of the variable, or want to place certain constraints on the trajectories of the ODEs. We leave this for future works.

## D   THEORETICAL ANALYSIS

We present the theoretical analysis for rectified flow. The results are summarized as follows.

• [Section D.1] All nonlinear rectified flows with any interpolation $X_t$ preserve the marginal laws.

• [Section D.2] The rectified flow (with the canonical linear interpolation) reduces convex transport costs.

• [Section D.3] Reflow guarantees to straighten the (linear) rectified flows.

• [Section D.4] We clarify the relation between straight couplings and $c$-optimal couplings.

## D.1   THE MARGINAL PRESERVING PROPERTY

The marginal preserving property that $\mathrm{Law}(Z_t) = \mathrm{Law}(X_t)$ for $\forall t$ is a general property of the nonlinear rectified flows in (8), regardless whether the interpolation $X_t$ is straight or not.

**Definition D.1.** *For a path-wise continuously differentiable random process $\boldsymbol{X} = \{X_t\colon t \in [0, 1]\}$, its expected velocity $v^{\boldsymbol{X}}$ is defined as*

$$v^{\boldsymbol{X}}(x, t) = \mathbb{E}[\dot{X}_t \mid X_t = x], \quad \forall x \in \mathrm{supp}(X_t).$$

*For $x \notin \mathrm{supp}(X_t)$, the conditional expectation is not defined and we set $v^{\boldsymbol{X}}$ arbitrarily, say $v^{\boldsymbol{X}}(x, t) = 0$.*

**Definition D.2.** *We call that $\boldsymbol{X}$ is rectifiable if $v^{\boldsymbol{X}}$ is locally bounded and the solution of the integral equation below exists and is unique:*

$$Z_t = Z_0 + \int_0^t v^{\boldsymbol{X}}(Z_t, t)\mathrm{d}t, \quad \forall t \in [0, 1], \quad Z_0 = X_0. \tag{11}$$

*In this case, $\boldsymbol{Z} = \{Z_t\colon t \in [0, 1]\}$ is called the rectified flow induced from $\boldsymbol{X}$.*

**Theorem D.3.** *Assume $\boldsymbol{X}$ is rectifiable and $\boldsymbol{Z}$ is its rectified flow. Then $\mathrm{Law}(Z_t) = \mathrm{Law}(X_t)$ for $\forall t \in [0, 1]$.*

*Proof.* For any compactly supported continuously differentiable test function $h\colon \mathbb{R}^d \to \mathbb{R}$, we have

$$\frac{\mathrm{d}}{\mathrm{d}t}\mathbb{E}[h(X_t)] = \mathbb{E}[\nabla h(X_t)^\top \dot{X}_t] = \mathbb{E}[\nabla h(X_t)^\top v^{\boldsymbol{X}}(X_t, t)], \tag{12}$$

where we used $v^{\boldsymbol{X}}(X_t, t) = \mathbb{E}[\dot{X}_t | X_t]$. By definition, this is equivalent to that $\pi_t := \mathrm{Law}(X_t)$ solves in the sense of distributions the continuity equation with drift $v_t^{\boldsymbol{X}} := v^{\boldsymbol{X}}(\cdot, t)$:

$$\dot{\pi}_t + \nabla \cdot (v_t^{\boldsymbol{X}} \pi_t) = 0. \tag{13}$$

To see the equivalence of (12) and (13), we can multiply (13) with $h$ and integrate both sides:

$$0 = \int h(\dot{\pi}_t + \nabla \cdot (v_t^{\boldsymbol{X}} \pi_t)) = \int h\dot{\pi}_t - \nabla h^\top v_t^{\boldsymbol{X}} \pi_t = \frac{\mathrm{d}}{\mathrm{d}t}\mathbb{E}[h(X_t)] - \mathbb{E}[\nabla h(X_t)^\top v^{\boldsymbol{X}}(X_t, t)],$$

where we use integration by parts that $\int h\nabla \cdot (v_t^{\boldsymbol{X}} \pi_t) = -\int \nabla h^\top (v_t^{\boldsymbol{X}} \pi_t)$.

Because $Z_t$ is driven by the same velocity field $v^{\boldsymbol{X}}$, its marginal law $\mathrm{Law}(Z_t)$ solves the very same equation with the same initial condition ($Z_0 = X_0$). Hence, the equivalence of $\mathrm{Law}(Z_t)$ and $\mathrm{Law}(X_t)$ follows if the solution of (13) is unique, which is equivalent to the uniqueness of the solution of $\mathrm{d}Z_t = v^{\boldsymbol{X}}(Z_t, t)$ following Corollary 1.3 of Kurtz (2011) (see also Theorem 4.1 of Ambrosio & Crippa (2008)). $\qquad\square$

## D.2   REDUCING CONVEX TRANSPORT COSTS

The fact that $(Z_0, Z_1)$ yields no larger convex transport costs than $(X_0, X_1)$ is a consequence of using the special linear interpolation $X_t = tX_1 + (1 - t)X_0$ as the geodesic of Euclidean space.

**Definition D.4.** *A coupling $(X_0, X_1)$ is called rectifiable if its linear interpolation process $\boldsymbol{X} = \{tX_1 + (1 - t)X_0\colon t \in [0, 1]\}$ is rectifiable. In this case, the $\boldsymbol{Z} = \{Z_t\colon t \in [0, 1]\}$ in (11) is called the rectified flow of coupling $(X_0, X_1)$, denoted as $\boldsymbol{Z} = \mathrm{RectFlow}((X_0, X_1))$, and $(Z_0, Z_1)$ is called the rectified coupling of $(X_0, X_1)$, denoted as $(Z_0, Z_1) = \mathrm{RectFlow}((X_0, X_1))$.*

**Theorem D.5.** *Assume $(X_0, X_1)$ is rectifiable and $(Z_0, Z_1) = \mathrm{RectFlow}((X_0, X_1))$. Then for any convex function $c\colon \mathbb{R}^d \to \mathbb{R}$, we have*

$$\mathbb{E}[c(Z_1 - Z_0)] \le \mathbb{E}[c(X_1 - X_0)].$$

*Proof.* The proof is based on elementary applications of Jensen's inequality.

$$
\begin{aligned}
\mathbb{E}\left[c(Z_1 - Z_0)\right] = \mathbb{E}\left[c\left(\int_0^1 v^{\boldsymbol{X}}(Z_t, t)\mathrm{d}t\right)\right] \quad &\text{//as } \mathrm{d}Z_t = v^{\boldsymbol{X}}(Z_t, t)\mathrm{d}t \\
\leq \mathbb{E}\left[\int_0^1 c\left(v^{\boldsymbol{X}}(Z_t, t)\right)\mathrm{d}t\right] \quad &\text{//convexity of } c, \text{ Jensen's inequality} \\
= \mathbb{E}\left[\int_0^1 c\left(v^{\boldsymbol{X}}(X_t, t)\right)\mathrm{d}t\right] \quad &\text{//} X_t \text{ and } Z_t \text{ shares the same marginals} \\
= \mathbb{E}\left[\int_0^1 c\left(\mathbb{E}\left[(X_1 - X_0) \mid X_t\right]\right)\mathrm{d}t\right] \quad &\text{//definition of } v^{\boldsymbol{X}} \\
\leq \mathbb{E}\left[\int_0^1 \mathbb{E}\left[c\left(X_1 - X_0\right) \mid X_t\right]\mathrm{d}t\right] \quad &\text{//convexity of } c, \text{ Jensen's inequality} \\
= \int_0^1 \mathbb{E}\left[c\left(X_1 - X_0\right)\right]\mathrm{d}t \quad &\text{//}\mathbb{E}[\mathbb{E}[(X_1 - X_0)|X_t]] = \mathbb{E}[(X_1 - X_0)] \\
= \mathbb{E}\left[c\left(X_1 - X_0\right)\right].
\end{aligned}
$$

$\square$

If $X_t$ is straight but with positive non-constant speed, that is, $X_t = \alpha_t X_1 + \beta_t X_0$ with $\beta_t = 1 - \alpha_t$ and $\dot{\alpha}_t \geq 0$, then we still have $\mathbb{E}[c(Z_1 - Z_0)] \leq \mathbb{E}[c(X_1 - X_0)]$ if $c$ is convex and $m$-homogeneous in that $c(ax) = |a|^m c(x)$ for $\forall a \in \mathbb{R}, x \in \mathbb{R}^d$, with some constant $m \in (0, 1]$.

### D.3 THE STRAIGHTENING EFFECT

A coupling $(X_0, X_1)$ is said to be straight (or fully rectified) if it is a fixed point of the $\texttt{RectFlow}(\cdot)$ mapping. It is desirable to obtain a straight coupling because its rectified flow is straight and hence can be simulated exactly with one step using numerical solvers. In this section, we first characterize the basic properties of straight couplings, showing that a coupling is straight iff its linear interpolation paths do not intersect with each other. Then, we prove that recursive rectification straightens the coupling and its related flow with a $O(1/k)$ rate, where $k$ is the number of rectification steps.

**Theorem D.6.** *Assume $(X_0, X_1)$ is rectifiable. Let $X_t = tX_1 + (1 - t)X_0$ and $\boldsymbol{Z} = \texttt{RectFlow}((X_0, X_1))$. Then $(X_0, X_1)$ is a straight coupling iff the following equivalent statements hold.*

1. *There exists a strictly convex function $c\colon \mathbb{R}^d \to \mathbb{R}$, such that $\mathbb{E}[c(Z_1 - Z_0)] = \mathbb{E}[c(X_1 - X_0)]$.*

2. *$(X_0, X_1)$ is a fixed point of $\texttt{RectFlow}(\cdot)$, that is, $(X_0, X_1) = (Z_0, Z_1)$.*

3. *The rectified flow coincides with the linear interpolation process: $\boldsymbol{X} = \boldsymbol{Z}$.*

4. *The paths of the linear interpolation $\boldsymbol{X}$ do not intersect:*

$$
V((X_0, X_1)) := \int_0^1 \mathbb{E}\left[\|X_1 - X_0 - \mathbb{E}\left[X_1 - X_0 \mid X_t\right]\|^2\right]\mathrm{d}t = 0, \tag{14}
$$

   *where $V((X_0, X_1)) = 0$ indicates that $X_1 - X_0 = \mathbb{E}[X_1 - X_0|X_t]$ almost surely when $t \sim Uniform([0, 1])$, meaning that the lines passing through each $X_t$ is unique, and hence no linear interpolation paths intersect.*

*Proof.* $3 \to 2 \to 1$: Obvious.

$1 \to 4$: If $\mathbb{E}[c(Z_1 - Z_0)] = \mathbb{E}[c(X_1 - X_0)]$, the two applications of Jensen's inequality in the proof of Theorem D.5 are tight. Because $c$ is strictly convex, the second Jensen's inequality in the proof implies that $X_1 - X_0 = \mathbb{E}[X_1 - X_0 \mid X_t]$ almost surely w.r.t. $\boldsymbol{X}$ and $t \sim \text{Uniform}([0, 1])$, which implies that $V(\boldsymbol{X}) = 0$.

$4 \to 3$: If $V(\boldsymbol{X}) = 0$, we have $\int_0^s (X_1 - X_0)\mathrm{d}t = \int_0^s \mathbb{E}[X_1 - X_0 | X_t]\mathrm{d}t = \int_0^s v^X(X_t, t)\mathrm{d}t$ for $s \in (0, 1]$. Hence

$$X_t = X_0 + \int_0^t (X_1 - X_0)\mathrm{d}t = X_0 + \int_0^t v^X(X_t, t)\mathrm{d}t.$$

Because $\boldsymbol{Z}$ satisfies the same equation (11), we have $\boldsymbol{X} = \boldsymbol{Z}$ by the uniqueness of the solution.

$\square$

$O(1/K)$ **convergence rate**    We now show that as we apply rectification recursively, the rectified flows become increasingly straight and the linear interpolation of the couplings becomes increasingly non-intersecting.

**Theorem D.7.** *Let $\boldsymbol{Z}^k$ the $k$-th rectified flow of $(X_0, X_1)$, that is, $\boldsymbol{Z}^{k+1} = \mathtt{RectFlow}((Z_0^k, Z_1^k))$ and $(Z_0^0, Z_1^0) = (X_0, X_1)$. Assume each $(Z_0^k, Z_1^k)$ is rectifiable for $k = 0, \ldots, K$.*

*Then*

$$\sum_{k=0}^{K} S(\boldsymbol{Z}^{k+1}) + V((Z_0^k, Z_1^k)) \leq \mathbb{E}\left[\|X_1 - X_0\|^2\right].$$

Hence, $\mathbb{E}[\|X_1 - X_0\|^2] < +\infty$, we have $\min_{k \leq K}(S(\boldsymbol{Z}^k) + V((Z_0^k, Z_1^k)) = O(1/K)$.

*Proof.* Taking $c(x) = \|x\|^2$ in the proof of Theorem 3.5, we can obtain that
$$\mathbb{E}\left[\|X_1 - X_0\|\right] - \mathbb{E}\left[\|Z_1 - Z_0\|\right] = S(\boldsymbol{Z}) + V((X_0, X_1)). \tag{15}$$
Applying it to each rectification step yields
$$\mathbb{E}\left[\|Z_1^k - Z_0^k\|^2\right] - \mathbb{E}\left[\|Z_1^{k+1} - Z_0^{k+1}\|^2\right] = S(\boldsymbol{Z}^{k+1}) + V((Z_0^k, Z_1^k)).$$
A telescoping sum on $k = 0, \ldots, K$ gives the result.

$\square$

### D.4    STRAIGHT VS. OPTIMAL COUPLINGS

A coupling $(X_0, X_1)$ is called $c$-optimal if it achieves the minimum of $\mathbb{E}[c(X_1 - X_0)]$ among all couplings that share the same marginals. Understanding and computing the optimal couplings have been the main focus of optimal transport (e.g., Villani, 2009; Ambrosio et al., 2021; Figalli & Glaudo, 2021; Peyré et al., 2019). Straight couplings is a different desirable property. In the following, we show that straightness is a necessary but not sufficient condition of being $c$-optimal for a strictly convex function $c$, except in the one dimensional case when the two concepts coincides. Hence, it is "easier" to find a straight coupling than a $c$-optimal couplings.

**Theorem D.8.** *If a rectifiable coupling $(X_0, X_1)$ is $c$-optimal for some strictly convex cost function $c$, then $(X_0, X_1)$ is a straight coupling.*

*Proof.* Let $(Z_0, Z_1) = \mathtt{RectFlow}((X_0, X_1))$. If $(X_0, X_1)$ is $c$-optimal, we must have $\mathbb{E}[c(Z_1 - Z_0)] = \mathbb{E}[c(X_1 - X_0)]$. This implies Statement 1 in Theorem D.6 and hence that $(X_0, X_1)$ is straight. $\square$

**1D Case**    For any $\pi_0, \pi_1$ on $\mathbb{R}$, there exists an unique coupling $(X_0^*, X_1^*)$ that is simultaneously optimal for all non-negative convex cost functions $c$. This coupling is uniquely characterized by a monotonic property: for every $(x_0, x_1)$ and $(x_0', x_1')$ in the support of $(X_0^*, X_1^*)$, if $x_0 < x_0'$, then $x_1 \leq x_1'$. Furthermore, if $\pi_0$ is absolutely continuously w.r.t. the Lebesgue measure, then $(X_0^*, X_1^*)$ must be deterministic in that there exists a mapping $T \colon \mathbb{R} \to \mathbb{R}$ such that $X_1^* = T(X_0^*)$. See Santambrogio (2015).

In the following, we show that straight couplings on $\mathbb{R}$ coincides with the deterministic monotonic coupling $(X_0^*, X_1^*)$ and hence is unique and simultaneously optimal for all convex $c$ when $\pi_0$ is absolutely continuous. The idea is that, in $\mathbb{R}$, a coupling is monotonic iff its linear interpolation paths do not intersect, a characteristic feature of straight couplings.

**Lemma D.9.** *A coupling on $\mathbb{R}$ is straight iff it is deterministic and monotonic.*

**Theorem D.10.** *For any $\pi_0, \pi_1$ on $\mathbb{R}$, there exists either no straight coupling, or a unique straight coupling. Further, if exists, the unique straight coupling is deterministic and monotonic, and jointly optimal w.r.t. all convex cost functions $c \colon \mathbb{R}^d \to [0, +\infty)$ for which the minimum value of $\mathbb{E}\left[c(X_1 - X_0)\right]$ exists and is finite.*

*Proof of Lemma D.9.* If $(X_0, X_1)$ on $\mathbb{R}$ is straight, then it coincides with its rectified coupling $(Z_0, Z_1) = \texttt{RectFlow}((X_0, X_1))$. But because $(Z_0, Z_1)$ is induced from the rectified flow $\mathrm{d}Z_t = v^X(Z_t, t)\mathrm{d}t$, it is obviously deterministic. It is also monotonic due to the non-crossing property of flows. Specifically, if $(Z_0, Z_1)$ is not monotonic, there exists $(z_0, z_1)$ and $(z_0', z_1')$ in the support of $(Z_0, Z_1)$ such that $z_0 < z_0'$ and $z_1 > z_1'$. If this happens, there must exists $t_0 \in (0, 1)$, such that $z_{t_0} = z_{t_0}'$. But by the uniqueness of the solution, we have $z_t = z_t$ for $t \geq t_0$, which is conflicting with $z_1 > z_1'$.

Assume $(X_0, X_1)$ is deterministic and monotonic. Due to the monotonicity, there exists no $x_0$ and $x_0'$ in the support of $\pi_0$, such that $x_0 \neq x_0'$ and $x_{t_0} = x_{t_0}'$ for some $t_0 < 1$. This suggests that $X_1 - X_0 = \mathbb{E}[X_1 - X_0 \mid X_t] = v^X(X_t, t)$ for $t \in (0, 1)$, and hence $\mathrm{d}X_t = (X_1 - X_0)\mathrm{d}t = v^X(X_t)\mathrm{d}t$, which is the ODE of the rectified flow. In addition, $X_t$ is obviously the unique solution of this ODE. Hence $(X_0, X_1)$ is rectifiable and straight following Statement 3 of Theorem D.6. □

*Proof of Theorem D.10.* This is the result of Lemma D.9 combined with the fact that the monotonic coupling is unique and jointly optimal for all convex $c$ for which the optimal coupling exists, following Lemma 2.8 and Theorem 2.9 of Santambrogio (2015). □

**Multi-dimensional cases** On the other hand, on $\mathbb{R}^d$ with $d \geq 2$, the different cost functions $c$ do not share a common optimal coupling in general, and a straight coupling is not guaranteed to optimize a specific $c$; this is expected because the $\texttt{RectFlow}(\cdot)$ procedure does not depend on a particular choice of $c$. Hence, one must modify the $\texttt{RectFlow}(\cdot)$ procedure to tailor it to a specific $c$ of interest.

In a recent work (Khrulkov & Oseledets, 2022), it was conjectured that the couplings $(Z_0, Z_1)$ induced from VP ODE (equivalently DDIM) yields an optimal coupling w.r.t. the quadratic loss, which was proved to be false in Lavenant & Santambrogio (2022); Tanana (2021). Here we show that even straight couplings are not guaranteed to be optimal, not to mention that VP ODE does not follow straight paths by design.

We explore this in a separate work (Liu, 2022b) that is devoted to modifying rectified flow to find $c$-optimal couplings, a result that can be easily stated is that the optimal coupling w.r.t. the quadratic cost $c(\cdot) = \|\cdot\|^2$ can be achieved as the fixed point of $\texttt{RectFlow}(\cdot)$ if $v$ is restricted to be a gradient field of form $v(x, t) = \nabla f(x, t)$ when solving (1). Restricting $v$ to be a gradient field removes the rotational component of the velocity field $v^X$ that causes sub-optimal transport cost.

## D.5 Additional Toy Examples

To accurately illustrate the theoretical properties, we use the non-parametric estimator $v^{X,h}(z, t)$ in (7) in the toy examples in Figure 2, 3, 11, 12. In practice, we approximate the expectation in (7) an nearest neighbor estimator: given a sample $\{x_0^{(i)}, x_1^{(i)}\}_i$ drawn from $(X_0, X_1)$, we estimate $v^X$ by

$$v^{X,h}(z, t) \approx \sum_{i \in \text{knn}(z,m)} \frac{x_1^{(i)} - z}{1 - t} \omega_h(x_t^{(i)}, z) \Big/ \sum_{i \in \text{knn}(z,m)} \omega_h(x_t^{(i)}, z), \quad x_t^{(i)} = t x_1^{(i)} + (1 - t) x_0^{(i)},$$

where $\text{knn}(z, m)$ denotes the top $m$ nearest neighbors of $z$ in $\{x_t^{(i)}\}_i$. We find that the results are not sensitive to the choice of $m$ and the bandwidth $h$ (see Figure 14). We use $h = 1$ and $m = 100$ by default. The flows are simulated using Euler method with a constant step size of $1/N$ for $N$ steps. We use $N = 100$ steps unless otherwise specified.

Alternatively, $v^X$ can be parameterized as a neural network and trained with stochastic gradient descent or Adam. Figure 14 shows an example of when $v^X$ is parameterized as an 2-hidden-layer fully connected neural network with 64 neurons in both hidden layers. We see that the neural networks fit

less perfectly with the linear interpolation trajectories (which should be piece-wise linear in this toy example). As shown in Figure 14, we find that enhancing the smoothness of the neural networks (by increasing the L2 regularization coefficient during training) can help straighten the flow, in addition to the rectification effect.

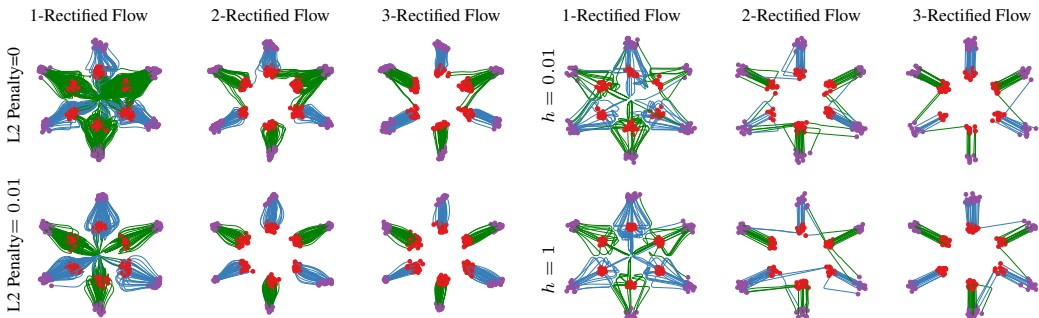

Figure 14: Rectified flows fitted with neural networks trained with different L2 penalty (left), and kernel estimator with different bandwidth $h$ (right). $\pi_0$: red dots; $\pi_1$: purple dots.

In Figure 3 of Section 2.2, the straightness is calculated as the empirical estimation of (3) based on the simulated trajectories. The relative transport cost is calculated based on $\{z_0^{(i)}, z_1^{(i)}\}_{i=1}^n$ drawn from $(Z_0, Z_1)$ by simulating the flow, as $\frac{1}{n} \sum_{i=1}^n \left\| z_1^{(i)} - z_0^{(i)} \right\|^2 - \left\| z_1^{(i^*)} - z_0^{(i)} \right\|^2$, where $z_1^{(i^*)}$ is the optimal L2 assignment of $z_0^{(i)}$ obtained by solving the discrete L2 optimal transport problem between $\{z_0^{(i)}\}$ and $\{z_1^{(i)}\}$. We should note that this metric is only useful in low dimensions, as it tends to be identically zero in high dimensional cases even $v^X$ is set to be a random neural network. This misleading phenomenon is what causes Khrulkov & Oseledets (2022) to make the false hypothesis that DDIM yields L2 optimal transport.

## E  ADDITIONAL EXPERIMENT DETAILS

**Experiment Configuration on CIFAR10**  We conduct unconditional image generation with the CIFAR-10 dataset (Krizhevsky et al., 2009). The resolution of the images are set to $32 \times 32$. For rectified flow, we adopt the same network structure as DDPM++ in (Song et al., 2020b). The training of the network is smoothed by exponential moving average as in (Song et al., 2020b), with a ratio of 0.999999. We adopt Adam (Kingma & Ba, 2014) optimizer with a learning rate of $2e - 4$ and a dropout rate of 0.15.

For reflow, we first generate 4 million pairs of $(z_0, z_1)$ to get a new dataset $D$, then fine-tune the $i$-rectified flow model for $300,000$ steps to get the $(i+1)$-rectified flow model. We further distill these rectified flow models for few-step generation. To get a $k$-step image generator from the $i$-rectified flow, we randomly sample $t \in \{0, 1/k, \cdots, (k-1)/k\}$ during fine-tuning, instead of randomly sampling $t \in [0, 1]$. Specifically, for $k = 1$, we replace the L2 loss function with the LPIPS similarity (Zhang et al., 2018) since it empirically brings better performance.

**Experiment Configuration on High-resolution Image Generation**  In high-resolution image generation, we adopt the NCSN++ network, following (Song et al., 2020b). The other configurations are kept the same as CIFAR10. We provide additional quantitative results in Table 2. Because the dataset only contains $< 6000$ images, FID is computed with 5000 images.

**Expreiment Configuration on Image-to-Image Translation**  In this experiment, we also adopt the same U-Net structure of DDPM++ Song et al. (2020b) for representing the drift $v^X$. We follow the procedure in Algorithm 1. For the purpose of generative modeling, we set $\pi_0$ to be one domain dataset and $\pi_1$ the other domain dataset. For optimization, we use AdamW (Loshchilov & Hutter, 2017) optimizer with $\beta$ $(0.9, 0.999)$, weight decay $0.1$ and dropout rate $0.1$. We train the model with a batch size of 4 for $1,000$ epochs. We further apply exponential moving average (EMA) optimizer with coefficient $0.9999$. We perform grid-search on the learning rate from $\{5 \times 10^{-4}, 2 \times 10^{-4}, 5 \times 10^{-5}, 2 \times 10^{-5}, 5 \times 10^{-6}\}$ and pick the model with the lowest training loss.

| Method | NFE($\downarrow$) | FID ($\downarrow$) |
|---|---|---|
| *ODE* | *One-Step Generation (Euler solver, N=1)* | |
| **1-Rectified Flow (+*Distill*)** | 1 | 227.82 (25.38) |
| **2-Rectified Flow (+*Distill*)** | 1 | 167.79 (28.60) |
| *ODE* | *Full Simulation (Runge–Kutta (RK45), Adaptive N)* | |
| **1-Rectified Flow** | 201 | 13.71 |
| **2-Rectified Flow** | 166 | 20.67 |

Table 2: FID on AFHQ-CAT dataset.

| Method | ERM | IRM | ARM | Mixup | MLDG | CORAL | Ours |
|---|---|---|---|---|---|---|---|
| OfficeHome | $66.5 \pm 0.3$ | $64.3 \pm 2.2$ | $64.8 \pm 0.3$ | $68.1 \pm 0.3$ | $66.8 \pm 0.6$ | $68.7 \pm 0.3$ | $\mathbf{69.2 \pm 0.5}$ |
| DomainNet | $40.9 \pm 0.1$ | $33.9 \pm 2.8$ | $35.5 \pm 0.2$ | $39.2 \pm 0.1$ | $41.2 \pm 0.1$ | $\mathbf{41.5 \pm 0.2}$ | $\mathbf{41.4 \pm 0.1}$ |

Table 3: The accuracy of the transferred testing data using different methods, on the OfficeHome and Domain-Net dataset. Higher accuracy means the better performance.

We use the AFHQ (Choi et al., 2020), MetFace (Karras et al., 2020) and CelebA-HQ (Karras et al., 2018) dataset. Animal Faces HQ (AFHQ) is an animal-face dataset consisting of 15,000 high-quality images at $512 \times 512$ resolution. The dataset includes three domains of cat, dog, and wild animals, each providing 5000 images. MetFace consists of 1,336 high-quality PNG human-face images at $1024 \times 1024$ resolution, extracted from works of art. CelebA-HQ is a human-face dataset which consists of 30,000 images at $1024 \times 1024$ resolution. We randomly select $80\%$ as the training data and regard the rest as the test data, and resize the image to $512 \times 512$.

### E.1 DOMAIN ADAPTATION

A key challenge of applying machine learning to real-world problems is the domain shift between the training and test datasets: the performance of machine learning models may degrade significantly when tested on a novel domain different from the training set. Rectified flow can be applied to transfer the novel domain ($\pi_0$) to the training domain ($\pi_1$) to mitigate the impact of domain shift.

**Experiment settings** We test the rectified flow for domain adaptation on a number of datasets. DomainNet (Peng et al., 2019) is a dataset of common objects in six different domain taken from DomainBed (Gulrajani & Lopez-Paz, 2020). All domains from DomainNet include 345 categories (classes) of objects such as Bracelet, plane, bird and cello. Office-Home (Venkateswara et al., 2017) is a benchmark dataset for domain adaptation which contains 4 domains where each domain consists of 65 categories. To apply our method, first we map both the training and testing data to the latent representation from final hidden layer of the pre-trained model, and construct the rectified flow on the latent representation. We use the same DDPM++ model architecture for training. For inference, we set the number of steps of our flow model as 100 using uniform discretization. The methods are evaluated by the prediction accuracy of the transferred testing data on the classification model trained on the training data.

**Experiment Configuration** For training the model, we apply AdamW (Loshchilov & Hutter, 2017) optimizer with batch size 16, number of iterations 50k, learning rate $10^{-4}$, weight decay 0.1 and OneCycle (Smith & Topin, 2019) learning rate schedule.

**Results** As demonstrated in Table 3, the 1-rectified flow shows state-of-the-art performance on both DomainNet and OfficeHome. It is better or on par with the previous best approach (Deep CORAL (Sun & Saenko, 2016)), while sustainably improve over all other methods.

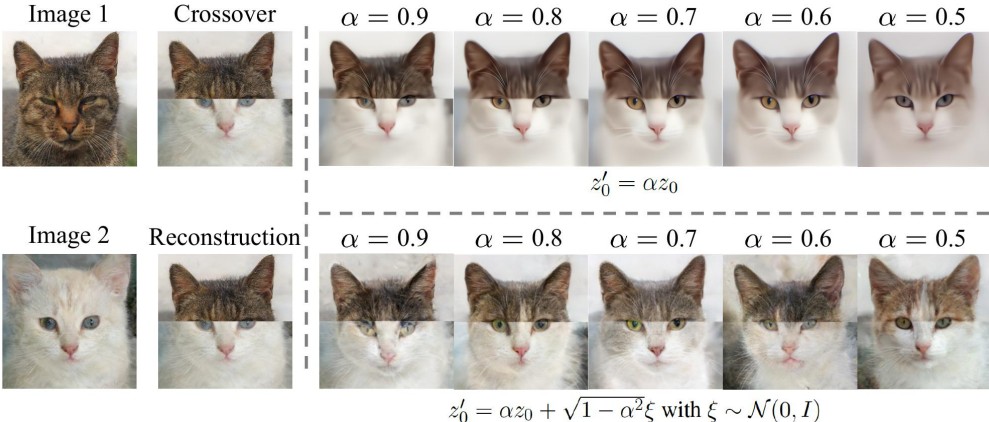

Figure 15: An example of image editing using 1-rectified flow. Here, we stitch the images of a white cat and a black cat into an unnatural image (denoted as $z_1$). We simulate the ODE reversely from $z_1$ to get the latent code $z_0$. Because $z_1$ is not a natural image, $z_0$ should have low likelihood under $\pi_0 = \mathcal{N}(0, I)$. Hence, we move $z_0$ towards the high probability region of $\pi_0$ to get $z_0'$ and solve the ODE forwardly to get a more realistically looking image $z_1'$. The modification can be done deterministically by improving the $\pi_0$-likelihood via $z_0' = \alpha z_0$ with $\alpha \in (0, 1)$, or stochastically by Langevin dynamics, which yields a formula of $z_0' = \alpha z_0 + \sqrt{1 - \alpha^2}\xi$ with $\xi \sim \mathcal{N}(0, I)$.

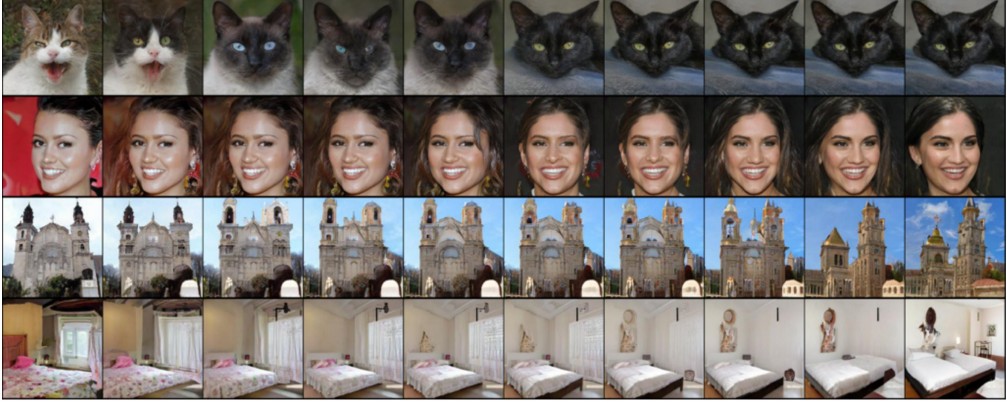

Figure 16: To visualize the latent space, we randomly sample $z_0$ and $z_1$ from $\mathcal{N}(0, I)$, and show the generated images of $\sqrt{\alpha}z_0 + \sqrt{1 - \alpha}z_1$ for $\alpha \in [0, 1]$.

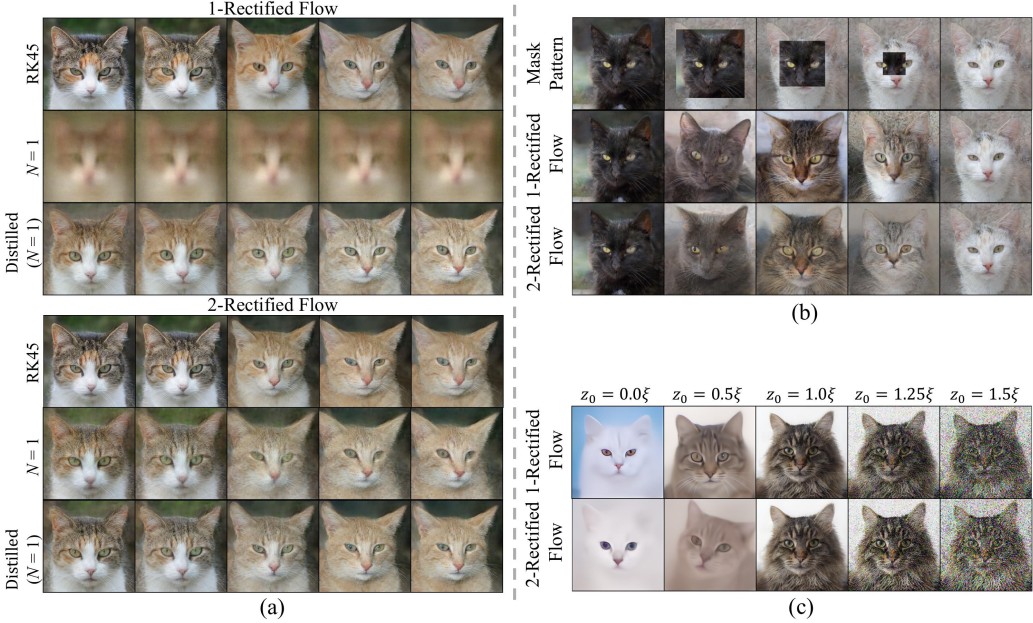

Figure 17: (a) We compare the latent space between Rectified Flow (0) and (1) using different sampling strategies with the same random seeds. We observe that (i) both 1-Rectified Flow and 2-Rectified Flow can provide a smooth latent interpolation, and their latent spaces look similar; (ii) when using one-step sampling ($N = 1$), 2-Rectified Flow can still provide visually recognizable interpolation, while 1-Rectified Flow cannot; (iii) Distilled one-step models can also continuously interpolate between the images, and their latent spaces have little difference with the original flow. (b) We composite the latent codes of two images by replacing the boundary of a black cat with a white cat, then visualize the variation along the trajectory. The black cat turns into a grey cat at first, then a cat with mixing colors, and finally a white cat. (c) We randomly sample $\xi \sim \mathcal{N}(0, I)$, then generate images with $\alpha\xi$ to examine the influence of $\alpha$ on the generated images. We find $\alpha < 1$ results in overly smooth images, while $\alpha > 1$ leads to noisy images.

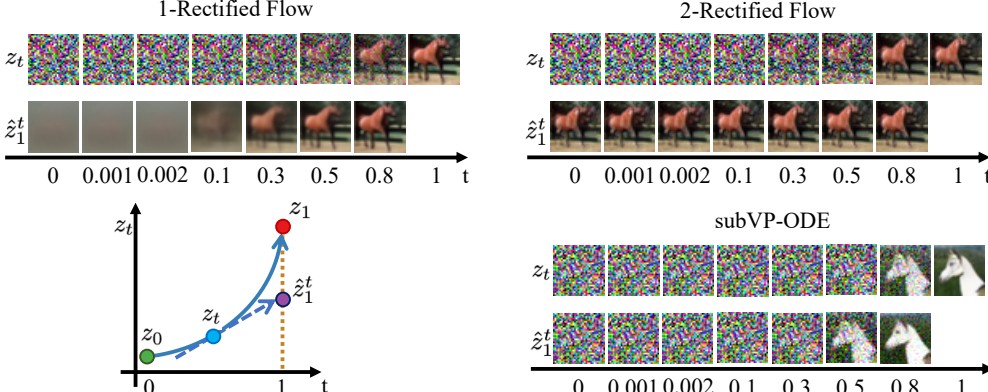

Figure 18: Sample trajectories $z_t$ of different flows on the CIFAR10 dataset, and the extrapolation $\hat{z}_1^t = z_t + (1 - t)v(z_t, t)$ from different $z_t$. The same random seed is adopted for all three methods. The $\hat{z}_1^t$ of 2-rectified flow is almost independent with $t$, indicating that its trajectory is almost straight.

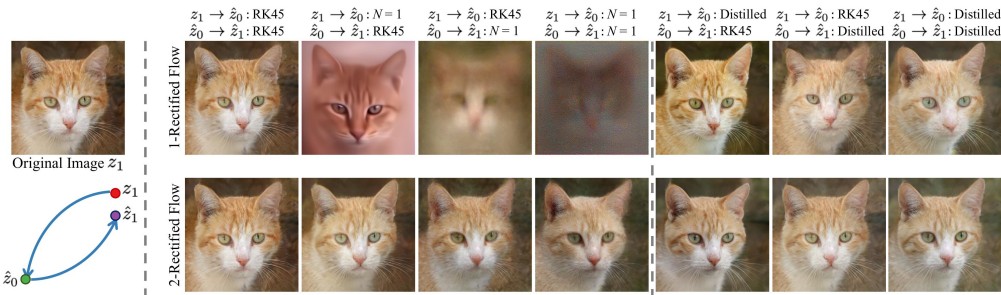

Figure 19: We perform latent space embedding / image reconstruction here. Given an image $z_1$, we use an *reverse ODE solver* to get a latent code $\hat{z}_0$, then use a *forward ODE solver* to get a reconstruction $\hat{z}_1$ of the image. The columns in the figure are *reverse ODE solver (forward ODE solver)*. (i) Thanks to the 'straightening' effect, 2-rectified flow can get meaningful latent code with only one reverse step. It can also generate recognizable images using one forward step. (ii) With the help of distilled models, one-step embedding and reconstruction is significantly improved.

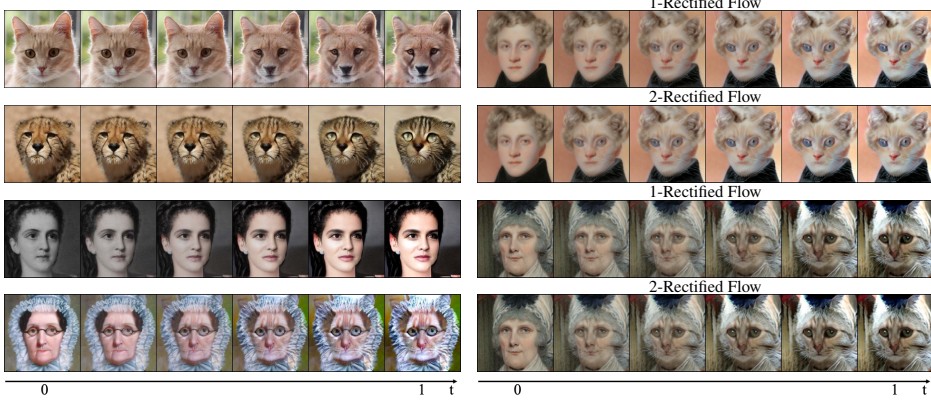

(a) 1-rectified flow between different domains

(b) 1- and 2-rectified flow for MetFace → Cat.

Figure 20: (a) Samples of trajectories $z_t$ of 1- and 2-rectified flow for transferring between different domains.

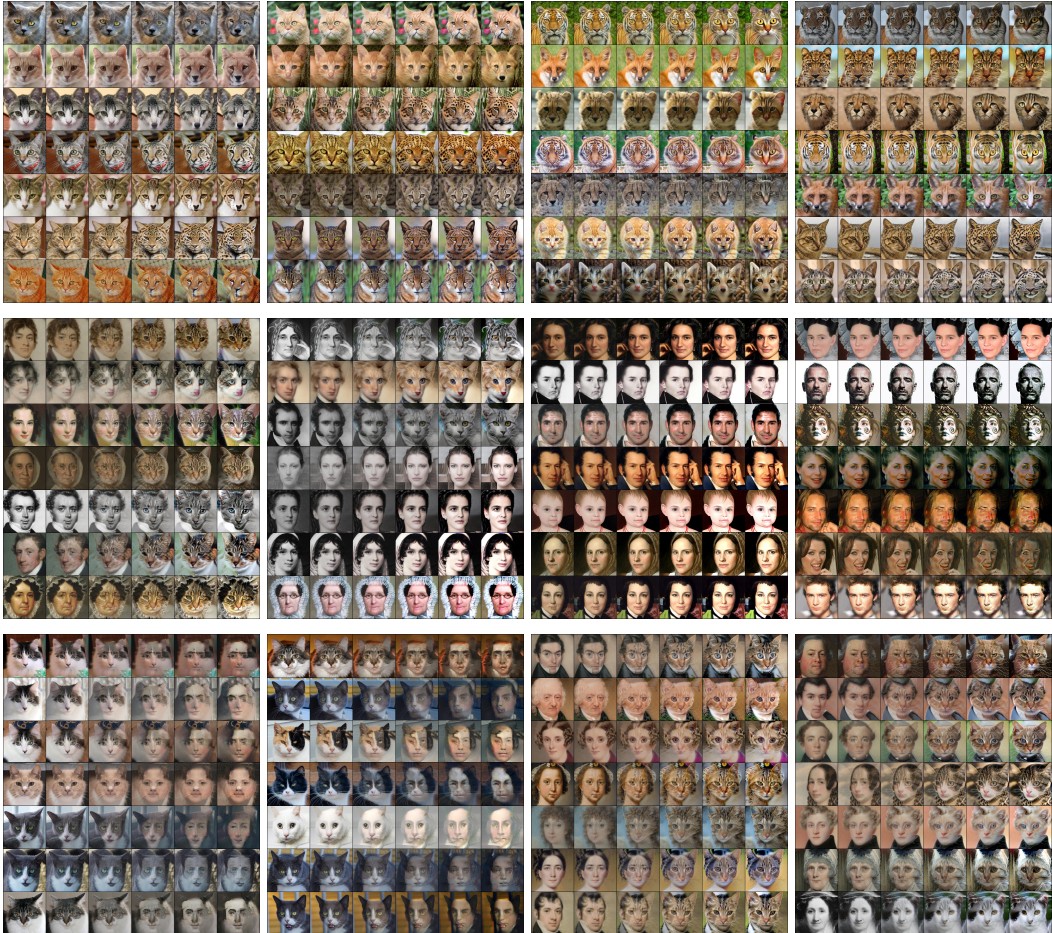

Figure 21: More results for image-to-image translation between different domains. The images in each row are time-uniformly sampled from the trajectory of 1-rectified flow solved $N = 100$ Euler steps with constant step size.

