# OpenReview forum: "Flow Straight and Fast: Learning to Generate and Transfer Data with Rectified Flow"
_ICLR.cc/2023/Conference — ICLR 2023 notable top 25%_

### Official Review · Reviewer_T1Pp · 2022-10-23

**Confidence:** 4
**Correctness:** 4
**Technical Novelty And Significance:** 3
**Empirical Novelty And Significance:** 3
**Recommendation:** 6

**Clarity, Quality, Novelty And Reproducibility:**

The paper is well written, although I think some points need to be explained more carefully. The idea is novel, and the authors give detailed algorithm for reproduction.

**Strength And Weaknesses:**

Strength:

1. The idea is simple but effective. The proposed ODE formulation is very simple, by learning the direction of linear interpolation between samples from two distribution along the interpolation path. However, experimental results show that this method is very effective.

2. It provides a new perspective of diffusion based models which are currently popular. The formulation itself is highly related to diffusion models. Actually, in the VP-SDE's formulation, the forward SDE looks like a spherical interpolation of data and noise. Here the model is doing linear interpolation, and the formulation only requires an ODE. Therefore, it shed new lights on studying diffusion models.

3. It provides a unified framework for generation and domain translation, which is an important contribution.

Weakness:
1. Although the ODE formulation and training objective is simple, I do not have a good intuitive understanding of choosing this ODE formulation. I found the sentence "To understand the method intuitively..." at the end of page 3 not helpful at all. A detailed explanation of why choosing this ODE formulation is necessary in the main text.

2. In addition, there seems not to have clear explanation on why reflow will make the flow more straight. It is not trivial to understand this.

2. Since one major selling point of this method is that it takes less NFEs to sample from, a more comprehensive comparison between existing related approaches on diffusion models is needed. For example, in table 1, I think results of DDIM (not the 1-step distillation) and progressive distillation [1] should be included. These results will better show the trade-off between NFE and FID. In particular, progressive distillation also does very well when NFE is low. Since distillation is also an (optional) part of your method, the comparison is even more important.

[1] Progressive Distillation for Fast Sampling of Diffusion Models, https://arxiv.org/abs/2202.00512

**Summary Of The Paper:**

This paper introduces rectified flow, which is an ODE model that connects two distributions and samples can be drawn by solving the ODE in two directions. In particular, the ODE aims to model the linear interpolation between samples from two distributions. Techniques like iteratively learning more ODEs and distillation are also introduced. Compared to SDE/ODE of diffusion based models, rectified flow has more linear paths, making numerically solving the ODE and sampling from it easier. Results show some improvements over diffusion based models, and its capability of image-to-image translation.

**Summary Of The Review:**

I think this paper would be a good contribution to the community. It has some issues in explaining the idea in an intuitive way, i.e., how the formulation is came up with, and I will increase the score if the presentation become better.

---

> ### Author Response · Authors · 2022-11-16
> **Response to Reviewer T1Pp**
>
> Thank you for your positive evaluation!
>
> **Q1: Intuitive Explanation.**
>
> Please see General Response \#1
> for an additional explanation of the method. We will incorporate them and further improve the draft based on reviewers' feedback.
>
> Meanwhile, we want to make a point that the method presented in this work is different from the many existing methods that can be  derived from an existing principle, such as maximum likelihood, or a standard optimization formulation.
> Instead, the work  follows a "here is the algorithm + here is the theorem on why the algorithm works" paradigm.
> Hence, we only have an intuitive explanation of the algorithm, rather than a logical derivation.
> In fact, we have seen different ways of deriving denoising diffusion models such as these the variational inference approach and the time-reversed SDE approach.
> One of the key observations that we had is that these derivations are in fact unnecessary and only obsolete the understanding of the algorithm and its design space.
> We believe that
> it is a better approach to
> directly accept the (very simple) algorithm as-is, combined with an intuitive understanding (the crossing avoiding intuition), and the rigorous theoretical analysis of its key properties we presented in the paper.
>
> **Q2: Why Does Reflow straighten flows?**
>
> At the intuitive level, it is due to the crossing avoiding property of flows. The rectified flow would be straight if all the linear interpolations of the coupling $(X_0,X_1)$  does not interact.
> On the other hand,
> due to the non-crossing property of flows,
> the rectification process reduces the change of having intersecting linear interpolations, and hence make the corresponding flow straighter.
> Please see General Response \#1 and the linked figure therein for more intuition.
>
> As you correctly observed, it is a new and non-trivial mathematical fact as made precise in Theorem D.7 that the straightness decays with an $ \min_{k\leq K} S(Z^k) = O(1/K)$ rate.
> The fact that we can only ensure the monotonic decrease of $\min_k S(Z^k)$, rather than $S(Z^k)$ adds another complication.
> Ultimately, any explanation is an incomplete explanation of the mathematical fact.
>
> **Q3: Comparison with DDIM and Progressive Distillation.**
>
> We've already provided comparison with VP-ODE in the paper, which is the continuous-time version of DDIM. We have added the progressive distillation in Table 1,
> which yields an FID of 9.12 compared with our 4.85 in the one step case.
> The trade-off between NFE and FID of rectified flow has been provided in Figure 4.

---

### Official Review · Reviewer_ot8j · 2022-10-25

**Confidence:** 3
**Correctness:** 3
**Technical Novelty And Significance:** 3
**Empirical Novelty And Significance:** 2
**Recommendation:** 8

**Clarity, Quality, Novelty And Reproducibility:**

The ideas presented in the paper are novel and creative. Details regarding the method and the setup used for the experiments are very extensive.
The paper has an odd structure. It only consists of 3 sections and does not have a conclusion section. I would advise the authors to add this section, as well as one discussing the background and related work.

**Strength And Weaknesses:**

### Strengths

The idea of transporting samples of one empirical distribution to another with a continuous normalizing flow is novel and creative. The learning algorithm is intuitive and thoroughly analyzed. Although being counterintuitive at first glance, the method is useful for many applications, the most obvious being domain adaptation like image-to-image translation, but also image generation from noise with improved sampling efficiency.

### Weaknesses

In my point of view, the most interesting application of this method would be domain adaptation, e.g. changing the weather or light conditions of the images captured by a (self-driving) car. This application is not explored in the paper. Instead, the authors focus on translating humans to cats, which might be amusing but not very useful.
Furthermore, the structure of the paper is poor. I elaborate on this below.


**Summary Of The Paper:**

In the paper, the authors introduce a method to train a continuous normalizing flow connecting two empirical distributions. They call their flow models rectified flows. These flows are trained by minimizing a simple least square objective given pairs of samples. Thereby, the flow is forced to approximately linearly interpolate between the samples. By iteratively retraining the model (reflow) the rectified flow learns to connect corresponding areas in the distribution.
The method is applied to several image generation benchmarks. The quality of the generated images is competitive with other procedures while being very efficient in sampling.

**Summary Of The Review:**

Overall, I tend towards accepting this paper, but ask the authors to restructure their paper and put a larger emphasis on their method being useful for domain adaptation.

Edit after rebuttal:

I appreciate the author's effort to improve the readability and theoretical soundness of their work. Therefore, I am in favor of accepting the article and raise my score.

---

> ### Author Response · Authors · 2022-11-16
> **Response to Reviewer ot8j**
>
> Thank you for your positive evaluation!
>
> **Q1: Application to domain adaptation.**
>
> We did apply our method in domain adaption in the DomainNet as shown in Appendix E.1. It was moved to appendix due to the page limit. We can add it to the main text when we have space. The self-driving car application is very interesting, valuable but challenging. We hope to see the application of our method on this problem in the future either by ourselves or by other researchers.
>
> Meanwhile, we think  the image translation task provides a nice way to demonstrate/visualize the power of our method on transferring complex and structure data. It also have direct applications on image editing and AI-based art generation.
>
> **Q2: The structure of the paper.**
>
> This work is comprehensive, and hence we decided to put the heavy theoretical content into appendix. We had a hard time to decide what goes to the paper and what goes to the appendix. We discussed background and related works in Section 1, and more thorough discussion of related works is found in Appendix A, and the other parts of Appendix.
> We have added the conclusion section in the current revised paper.

---

### Official Review · Reviewer_e8Br · 2022-10-25

**Confidence:** 4
**Correctness:** 4
**Technical Novelty And Significance:** 3
**Empirical Novelty And Significance:** 2
**Recommendation:** 6

**Clarity, Quality, Novelty And Reproducibility:**

Novelty:
  - To the best of my knowledge, this work is novel.
  - Getting good samples with N=1 is interesting, although these are purely qualitative and it's not clear how this compares to directly training a one-step model.

Quality:
  - The paper does a good job of explaining the algorithm since it is quite simple, but I had a hard time understanding the theoretical motivations for this objective.
  - Relatedly, as the paper focuses on sample quality comparisons on non-CIFAR10 data sets, it might make sense to have comparisons to GAN-based methods for training OT maps, i.e. works that directly learn the one-step transport map instead of straightening an ODE, e.g. [1,2]. The second reference has quantitative experiments on image-to-image translation task such as colorization, inpainting, etc. While GANs' optimization is adversarial, this does not necessarily mean that rectification is faster or more stable to train in practice, since it requires iterative rectification. IMO showing a bit more how Rectified Flows' iterative procedure is better than directly modeling OT solutions could make a nice addition to the paper.

Clarity:
  - Figure 2 is very nice.
  - The random variables for each expectation should be explicitly written out. The lack of clarity regarding this made the proofs hard to follow for me and often required some imagination. For example, Eq (12):  d/dt E[h(X_t)] = E[\nabla h(X_t) dX_t/dt ]. The first expression I assume is an expectation over X_t since this is the only random variable, but the second expression has two random variables (X_t and d/dt X_t). Is this equality the result of some stochastic calculus rule? It would be good to cite why this equality is true.
  - What are the numbers in parentheses in Figure 1(a) ?
  - The theoretical results assume that the random process is "path-wise continuously differentiable" in order to define stochastic derivatives. This seems like a rather strong assumption and standard SDEs would not satisfy this. Can the authors explain whether this is a restrictive assumption?

[1] "Optimal transport mapping via input convex neural networks." Makkuva et al. (2021)
[2] "Generative modeling with optimal transport maps." Rout et al. (2021)

**Strength And Weaknesses:**

Strengths:
  - The objective is simple and achieves reasonable results.

Weaknesses:
  - The theoretical details of the paper are difficult to follow. It would be good to have a clearer connection/derivation between the theoretical motivation (straight paths) and the objective itself.
  - Only image quality metrics are reported. Existing ODE/SDE models tend to report NLL in addition to FID, and it would be good to have NLL comparisons. This reviewer's experience with FID has been that it is a rather fickle metric, and FID on CIFAR-10 only compares to training data with no test for generalization.

**Summary Of The Paper:**

This paper proposes to learn continuous-time transport maps defined using ODEs to map between two distributions. The algorithm aims to regularize for straightness of the paths, which can help make simulation faster. The idea of Rectified Flow is to sample pairs (x0, x1) and fit a vector field to the linear interpolation between x0 and x1. This optimization process is then repeated by creating pairs of samples from the model.

The authors make a series of theoretical statements regarding this training process, especially that if this process is repeated K->inf times, then the paths become straight.

The experimental results show that the model is competitive in terms of FID with previous diffusion SDE models while being better than deterministic ODE counterparts.

**Summary Of The Review:**

This paper makes a number of interesting contributions that I think are useful for the ICLR community, so I vote for acceptance. I do think the writing can be made clearer, especially the connection between the theoretical details and the actual algorithm. How this explicitly relates to OT is also not completely clear, and is only hinted at in scattered places in the paper. Empirical results feel mainly qualitative, and I think (i) reporting NLL and (ii) doing more quantitative comparisons with existing models like GANs will make the results better.

---

> ### Author Response · Authors · 2022-11-16
> **Response to Reviewer e8Br**
>
> Thank you for your positive evaluation!
>
> **Q1: The theoretical details of the paper are difficult to follow. It would be good to have a clearer connection/derivation between the theoretical motivation (straight paths) and the objective itself.**
>
> We want to emphasize that the nature of the work is pretty different from typical methods that can be derived from an established principle like maximum likelihood, or a direct optimization of a given objective function.
> Strictly speaking, what we can provide is only
> an intuition explanation of the method, rather than a derivation from a familiar algorithm/framework. The validity of the method is justified by the three key theorems in Section 2.2.
> Especially when considering its connection to optimal transport, our framework is different in nature from any existing frameworks that is familiar in machine learning.
>
> Going back to your specific question, as already stated in General Response #1, our method **CANNOT** be treated as a direct optimization procedure of minimizing the straightness objective function, which would be something like
> $$
> \min_{\boldsymbol{Z}} S( \boldsymbol{Z}) ~~~s.t.~~~ \text{Law}(Z_0)=\pi_0,~~~~ \text{Law}(Z_1) = \pi_1.
> $$
> However, this formulation yields a challenging constrained optimization problem similar to those in optimal transport, and one would have to resort to some minimax procedure with Lagrangian multiplier to solve it, yielding similar instability issues of GANs.
>
> The reflow procedure is a less obvious approach to achieve $S(\boldsymbol{Z})=0$,
> which has the critical advantage of reducing to a sequence of standard unconstrained optimization problems without resorting to constrained or minimax optimization.
> Mathematically, the reflow procedure can be viewed as jointly minimizing the transport cost $\mathrm{E}[c(Z_1-Z_0)]$ for all convex $c$, until it reaches a point where it is impossible to further decrease every cost simultaneously. The $S(\boldsymbol{Z}) =0$ can be intuitively viewed as some sort of stationary (or zero-gradient) condition of such simultaneous (or multi-objective) optimization problem.
>
> **Q2: Numerical Results on Negative Log-likelihood**
>
> Here are some additional results of NLL on the CIFAR10 test set, measured by the same pipeline in the Appendix D.2 of [SY] using RK45 solver.
>
> 1-Rectified Flow: 3.15
>
> 2-Rectified Flow: 3.20
>
> 3-Rectified Flow: 3.18
>
> Overall, NLL of rectified Flow is on par with VP-ODE, which is 3.16.
> We do not expect a clear advantage on NLL as it is not directly related to our method in both theory and algorithm.
>
> [SY] Score-Based Generative Modeling through Stochastic Differential Equations, https://arxiv.org/abs/2011.13456
>
> **Q3: Comparison with GAN-based methods for training OT mappings.**
>
> In the following link, we show a multi-modal 2D toy example on which we compare our method with a GAN-based algorithm of OT in paper [1,2] pointed out by the reviewer:
>
> https://anonymous.4open.science/r/iclr2023_rebuttal_rectified_flow-D882/figs/ot_comparison.jpg
>
> We can see that rectified flow smoothly fits the target distribution with a simple L2 loss function, while GANs have trouble in fitting each mode precisely.
>
> Moreover, we'd like to clarify that our method does not solve the optimal transport problem.
> Instead, it solves a multi-objective variant of OT w.r.t. the whole family of transport cost $\mathrm{E}[c(Z_1-Z_0)]$ for all convex functions $c$.
> The fixed point of reflow does not return a  optimal coupling, but rather a *straight coupling* a concept that we introduce and discuss  in Appendix D.4.
>
> We refer the reviewer to
> the "Connection to Optimal Transport" part of General Response #1.
>
> Practically speaking,
> compared with typical OT-based methods such as
> these in paper [1][2] pointed out by the reviewer,
> the main advantage of
> rectified flow is that
>
> 1) It reduces to a sequence of standard unconstrained optimization problem, and does not lead to a numerically unstable minimax formula that typical methods require.
>
> 2) It does nor require to use
> special neural network architectures like input convex networks that some existing OT methods
> (e.g., paper [2]) require.  The special requirement on network architectures make it difficult to achieve state of the art results on challenging problems.

---

> > ### Author Response · Authors · 2022-11-16
> > **Continued**
> >
> > **Q4: Clarity on equations, e.g., Eq. (12).**
> >
> > As we wrote at the bottom of page 3,
> > in the case of linear interpolation $X_t = t X_1 + (1-t) X_0$, the expectation is taken w.r.t. the randomness of $(X_0,X_1)$, and $X_t$ and $\dot X_t = d X_t / d t = X_1-X_0$ are viewed as a function of $(X_0,X_1)$ when taking the expectation.
> >
> > In the case of nonlinear rectified flow, the interpolation process $\mathrm{X} = \{X_t\}$ can be viewed as a stochastic process which is trajectory-wise differentiable. More specifically, as what is standard in stochastic process, the stochastic process $X_t$ can be viewed as a measurable function $X(t, \omega)$, where $\omega$ can be viewed as a "random seed" intuitively, and the expectation is taken w.r.t. $\omega$ (In the linear case we have $\omega= (X_0,X_1)$, but more generally we can have extra randomness besides that from the end points).
> >
> > **Q5: Comparison with Directly Training a One-step Model**
> >
> > We have provided the results of TDPM (T=1) and Denoising diffusion GAN (DDGAN) (T=1) in Table 1 (b), which are both one-step generative models. TDPM uses the improved DDPM U-Net, and DDGAN uses NCSN++ U-Net. Both of them are trained with the  GAN-style minimax procedures.
> > We think that TDPM (T=1) and DDGAN (T=1) are the best-known results for one-step generation with the U-Net generators that are comparable to these used in diffusion models, and believe that they serve as proper baselines for *directly training with one-step model*.
> >
> > As shown in Table 1 (b), rectified Flow (4.85) beats both methods on FID by a noticeable margin (8.91, 14.6).
> > In addition, training a GAN requires a lot of tuning due to the instability issue and sophisticated design on the discriminator.
> >
> > **Q6: What are the numbers in parentheses in Figure 1(a) ?**
> >
> > The $N$ in Figure 1 refers to the number of simulation steps using Euler discretization. When $N=1$, we in fact generate images with one step $Z_1 = Z_0 + v(Z_0, 0)$.
> > For the right most column with `Distilled ($N=1$)', we further distill the learned rectified flows into a one step model, rather than directly running a one step Euler step.
> > See more in the pipeline on page 6 and Appendix E.
> >
> > **Q7: The assumption on "path-wise continuously differentiable".**
> >
> > Using the trick in [SY],  we can convert between SDEs and ODEs while preserving the final outcome distributions.
> > Hence, the set of models derived from  differentiable interpolations are not more
> > restrictive than that derived using SDE-based interpolation.
> >
> > Moreover, focusing on differentiable interpolation substantially simplifies the derivation without referring to the more advanced Ito calculus machinery.
> > See Section D.5 for more discussion regarding this issue.
> >
> > [SY] Score-Based Generative Modeling through Stochastic Differential Equations, https://arxiv.org/abs/2011.13456

---

### Author Response · Authors · 2022-11-16
**Greetings and Response to the Reviewers**

We thank the reviewers for the comments and questions! For all the reviewers, please first read General response #1 for a quick overview of the intuition and related theory, then General Response #2 for a summary of contributions. After that, we address the individual concerns from each reviewer. We would appreciate any increase in rating if your concern is properly addressed.

Best regards,

Authors

---

### Author Response · Authors · 2022-11-16
**General Response #1: A Quick Overview of the Method and Theory**

In response to some of the common questions the reviewers had, we provide here a quick overview of the main intuition of the method and the related theory.

**Problem setting**

Given two unknown distributions $\pi_0$ and $\pi_1$ which are observed through the marginal observations $\lbrace x_{0,i} \rbrace_{i=1}^n \sim \pi_0$ and $\lbrace x_{1,i}\rbrace_{i=1}^n \sim \pi_1$. We want to find a velocity field $v(x,t)$, such that $d Z_t= v(Z_t,t ) d t$ transports $\pi_0$ to $\pi_1$, that is, we have $Z_1\sim \pi_1$ starting from $Z_0\sim \pi_0$ following the ODE.

**Method**

Rectified flow
works by **finding an ODE that fits the linear interpolation of points from $\pi_0$ and $\pi_1$ on marginal distributions**.
Assume that we observe $X_0\sim \pi_0$ and
$X_1 \sim \pi_1$. Let $X_t$ be the linear (or
geodesic) interpolation of $X_0$ and $X_1$:

$$
X_t = t X_1 + (1-t) X_0, ~~~~\forall  t\in[0,1].
$$

Observe that $X_t$ follows
a trivial ODE that already transfers $\pi_0$ to $\pi_1$: \begin{align}
\mathrm{d}X_t  = (X_1 - X_0) \mathrm{d}t,~~~~~~~~~~(1)
\end{align}

in which $X_t$ moves following the line
direction $(X_1-X_0)$ following a constant
direction and speed. See Figure (a) in the link here:
https://anonymous.4open.science/r/iclr2023_rebuttal_rectified_flow-D882.

However, this ODE is not very useful: it can not be simulated **causally**, because the update $X_t$ depends on the **final state** $X_1$, which is not available at time $t<1$. In
the linked Figure (a), the non-causality is reflected in the
crossing points of the trajectories (see the middle of the figure where blue and green lines cross). When multiple trajectories
intersect at a point $X_t$, the update direction is non-unique and hence can not casually forward in time.

Hence, we want to *causalize* the interpolation process $X_t$, by *projecting* it to the space of causally simulatable ODEs of form $\mathrm{d} Z_t = v(Z_t, t) \mathrm{d} t $. A natural way to achieve this is the L2 projection on the velocity field,
finding $v$ by minimizing the least squares loss with the line directions $X_1-X_0$:
\begin{align}
\min_{v} \int_0^1  \mathbb{E}\left [\lVert{(X_1-X_0) - v(X_t, t)}\rVert^2\right] \mathrm{d}t. ~~~~~~~~~(2)
\end{align}

Theoretically, the solution can be represented using
conditional expectation:

$$
v(z,t) = \mathbb{E}[X_1-X_0 |X_t=z], ~~~~~~~~~~~~(3)
$$

which is the average of the directions of the lines that pass through
point $z$ at time $t$.
We call the ODE with $v$ in Eq. (2) and  Eq. (3) the rectified form induced from $(X_0,X_1)$.
**The expectation is taken w.r.t. the randomness of $(X_0,X_1)$ while treating $X_t = t X_1 + (1-t) X_0$ as a function of $(X_0,X_1)$. In other words, it is taken over the joint probability of $(X_0, X_1)$.**
In practice, we approximate  the expectation $\mathbb{E}[\cdot]$  with empirical draws of
$(X_0,X_1)$. Specifically,
let $(x_{0,i}, x_{1,i})_{i=1}^n$ be a set of realizations of $(X_0,X_1)$, and augment each data point with (multiple copies of) random time $t_i \sim \mathrm{Uniform}([0,1])$. Then $v$ is estimated by

$$
\min_{v}
\sum_{i=1}^n ||x_{1,i} - x_{0,i} - v(t x_{1,i} + (1-t) x_{0,i}, t)||^2.
$$

As shown in
the linked Figure (b), the trajectories $Z_t$ of rectified flow
traces out the same density map as that of the
interpolation trajectories $X_t$, but are *rewired* on the intersecting
points to avoid the non-causality.
The validity of this method is justified by the following key properties:

**Key Property 1: Marginal Preserving**

>The ODE trajectories $Z_t$ and the
interpolation $X_t$ have the same marginal distributions, that is,
$$\mathrm{Law}(Z_t) = \mathrm{Law}(X_t), ~~~ \forall t\in[0,1].$$

As $(X_0,X_1)$ is obviously a coupling of $\pi_0$ and $\pi_1$ by definition,  $(Z_0,Z_1)$ also forms a coupling of $\pi_0$ and $\pi_1$.

**Key Property 2: Transport Cost**

>$(Z_0,Z_1)$ guarantees to yield no larger transport cost than
$(X_0,X_1)$ simultaneously for **all convex cost functions $c$**, that
is,
$$\mathbb{E}[c(Z_1-Z_0)] \leq \mathbb{E}[c(X_1-X_0)],~~~
\text{$\forall$ convex $c\colon \mathbb{R}^d\to \mathbb{R}$}.$$

The data pair $(X_0,X_1)$ can be an arbitrary coupling of $\pi_0$ and $\pi_1$,
typically independent (i.e., $(X_0,X_1)\sim \pi_0\times \pi_1$), obtained by randomly combining observations from $\pi_0$ and $\pi_1$. In comparison, the rectified coupling $(Z_0,Z_1)$ has a
deterministic dependency as it is constructed from an ODE model. Hence,
**rectified flow converts an arbitrary coupling into a deterministic
coupling with no larger convex transport costs.**

---

> ### Author Response · Authors · 2022-11-16
> **Continued**
>
> **Reflow: Fast Generation with Straight Flows**
>
> Denote the rectified flow $\boldsymbol Z = \{Z_t: t\in[0,1]\} $
> induced from $(X_0,X_1)$ by
> $\boldsymbol Z = \mathsf{Rectflow}((X_0,X_1))$.  Applying this operator recursively yields a
> sequence of rectified flows
>
> $$
> \boldsymbol Z^{k+1} = \mathsf{Rectflow}((Z_0^k, Z_1^k))
> $$
>
> with $(Z_0^0,Z_1^0)=(X_0,X_1)$, where $\boldsymbol Z^k$ is the $k$-th rectified
> flow
> induced from $(X_0,X_1)$.
> In practice, this can be implemented by drawing samples of $(Z_0^k, Z_1^k)$ from the $k$-th rectified flow, and using them to find the new flow by the training procedure above (with $(X_0,X_1)$ replaced by $(Z_0^k,Z_1^k)$).
>
> Besides decreasing transport cost shown above,
> this **reflow** procedure has the important effect of **straightening paths**
> of rectified flows: the paths of $\boldsymbol Z^k$ are increasingly straight as $k$ increases.
>
> **Key Properties 3: The Straightening Effect**
>
> > Measure the straightness of $\boldsymbol Z$ by
> $s(\boldsymbol{Z}) = \int_0^1 \mathbb{E}[\lVert Z_1-Z_0 - v(Z_t,t)\rVert^2]\mathrm{d}t$, such that
> $S(\boldsymbol Z) =0$ corresponds to straight paths. Then we have
> $\min_{k\leq K}S(\boldsymbol Z^k) = O(1/K)$.
>
> Flows with nearly straight paths bring a key computational advantage as they incur small time-discretization error in numerical simulation.
> Indeed, if an ODE $\mathrm{d}Z_t = v(Z_t,t) \mathrm{d}t$ has perfectly straight paths,
> we have
> $$
> Z_1 = Z_0 + v(Z_0, 0),
> $$
> meaning that the ODE can be solved
> exactly with **a single Euler step**, which addresses the very
> bottleneck of slow inference of ODE/SDE models.
> Hence, this reflow/straightening procedure provides a special way for training one-step generative models (like GAN/VAE),
> by leveraging ODEs as an intermediate step.
> For practical image generation, we find that it is sufficient to only reflow once.
>
> **The Nature of Reflow**
>
> An important aspect of the reflow procedure
> is that it **DOES NOT** correspond to a direct optimization of the straightness measure $S(\boldsymbol Z)$, yet it still ensures $S(\boldsymbol{Z}) = 0$ at the fixed point.
> Note that a straightforward direct optimization framework would be something like
> $$\min_{\boldsymbol{Z}} S(\boldsymbol{Z}) ~~~s.t.~~~ \text{Law}(Z_0)=\pi_0,~~~~ \text{Law}(Z_1) = \pi_1,
> $$
> which yields a challenging constrained optimization problem similar to these in optimal transport, and one would have to resort to some minimax procedure with Lagrangian multiplier to solve it, which has the instability issues of GANs.
>
>
> The reflow procedure is a less obvious approach to achieve $S(\boldsymbol{Z})=0$,
> which has the critical advantage of reducing to a sequence of standard unconstrained optimization problems without resorting to constrained or minimax optimization.
> Mathematically, the reflow procedure can be viewed as jointly minimizing the transport cost $mathrm{E}[c(Z_1-Z_0)]$ for all convex $c$, until it reaches a point where it is impossible to further decrease every cost simultaneously. The $S(\boldsymbol{Z}) =0$ can be intuitively viewed as some sort of stationary (or zero-gradient) condition of such simultaneous (or multi-objective) optimization problem.

---

> > ### Author Response · Authors · 2022-11-16
> > **Connection to Optimal Transport (OT): Multi vs. Single Objective**
> >
> > **The precise mathematical relation of diffusion/flow models and optimal transport (OT) is highly subtle and non-trivial.
> > We think a major contribution of our work is to clarify this relation precisely, which we believe is a deep and non-trivial mathematical contribution. We hope the reviewers can embrace the technical complexity here and appreciate the mathematical depth of this part of the work in additional to the empirical aspects.**
> >
> > Rectified flow is *aggressive* in that it attempts to decrease the
> > transport cost  $\mathrm{E}[c(Z_1-Z_0)]$
> >  simultaneously for **all convex functions $c$**, without preferring or
> > specifying any particular $c$.
> > Hence, it can be viewed as a variant of multi-objective optimization that attends to optimize the whole family of convex costs.
> > Because different costs $c$ may conflict with each other, our procedure can not guarantee to optimize all or any given $c$. This differentiates our method from the traditional OT that minimizes a single $\mathrm{E}[c(Z_1-Z_0)]$ with a pre-specified choice of $c$ (say $c(x)=\frac{1}{2}||x||^2$).
> >
> > Note that for generative and transfer modeling tasks considered in this problem, the transport cost is not a direct objective of interest. Hence, having to pre-specify and optimize a given $c$ in OT is unnecessary or even disadvantageous for the learning task.
> >
> > One the other hand, it is possible to modify
> > the rectified flow procedure to target the specific $c$  (say the canonical
> > quadratic cost $c(x) = \frac{1}{2} \lVert{x}\rVert^2$) that we are
> > interested in, perhaps with the cost of increasing the other costs. In fact, for the quadratic cost $c(x) = \frac{1}{2}\lVert{x}\rVert^2$, we simply need to
> > constraint the drift $v$ to be a gradient field of form $v(z,t) = \nabla_z f(z,t)$ during the optimization. As it is beyond the scope of this work, which focuses on generative/transfer modeling, rather than OT, the detailed discussion should be deferred to separate works.

---

### Author Response · Authors · 2022-11-16
**General Response #2: Summary of Contributions**

Here we provide a summary of our contribution and the advantages of our method.

* Rectified flow provides a simple and unified treatment for both generative and transfer modeling.

* The framework is purely ODE-based, avoiding the more mathematically sophisticated SDE models both conceptually and algorithmically.

* Our method simplifies the existing diffusion/flow models. It decouples and simplifies the design space of existing diffusion/flow models. It avoids some unnecessary parameters, decouples the choice of initial distributions, and shows that RF is in fact the canonical choice.

* With the novel **reflow** procedure, we propose learning straight flows as a  simple yet principled approach to fast inference in flow models. It effectively provides a new way for training one-step models with ODEs as intermediate steps.
Empirically, we are the first result that can generate high-quality high-resolution images with only a single Euler step in flow models, for both generative and transfer settings.

* The novel theoretical and algorithmic insights to optimal transport are of independent interest.
 We, for the first time, elaborate the precise connection between our method with optimal transport by showing rectification jointly decreases all convex transport costs.  The rectification is a new mathematical concept. It is a *universal* descender of all convex transport cost. It is also possible to modify the operator to optimize a specific transport cost.

* With non-linear rectified flow, we provide a new and simplified  way for understanding the popular diffusion models and their ODE variants.

* The intuitive idea of `causalizing interpolation processes` provides a new paradigm
for solving related distribution transfer problems in general, and is amenable to rigorous theoretical analysis.

* In terms of empirical results, as far as we know, rectified flow is the first to achieve a recall score of 0.51 and a FID of 5.21 simultaneously on CIFAR-10 with **one step**. It also beats the previous fast probabilistic flow methods by a large margin with one-step generation in FID (ours: 4.85, progressive distillation: 9.12, TDPM: 8.91, DDGAN: 14.6, DDIM Distillation: 9.36). Besides, as our framework provides a much freer choice for the initial distribution $\pi_0$, our framework significantly simplifies the design of domain transfer with diffusion/flow. In fact, our framework unifies domain transfer with generative modeling.
For a more thorough discussion about the relationship, improvements and differences, we kindly refer the reviewers to refer to Appendix A.

---

### Decision · Program_Chairs · 2023-01-20

**Decision:**

Accept: notable-top-25%

**Justification For Why Not Higher Score:**

It's an interesting paper but without substantial enough innovation or impressive enough results for an oral.

**Justification For Why Not Lower Score:**

Spotlight may make sense since the idea is quite elegant and conceptually nice, so could be highlighted.

**Metareview: Summary, Strengths And Weaknesses:**

The paper proposes an approach for learning to transport between two distributions, with applications to generative modeling and domain transfer (image-to-image translation).

The reviewers are all in favor of acceptance, with the following key arguments.

Pros:
- Simple, elegant, and new method
- Good empirical performance
- Interesting theoretical results

Cons:
- Non-ideal presentation: intuitions and theory are at times difficult to understand
- Somewhat limited experimental evaluation and comparison to baselines

I concur with the overall positive sentiment, especially given that the authors addressed some of the negative points quite convincingly in the rebuttal. I, therefore, recommend acceptance but urge the authors to ensure the final version addresses the reviewers' concerns as well as possible.

**Note From Pc:**

if the above contains the word "oral" or "spotlight" please see: "oral" presentation means -> notable-top-5% and "spotlight" means -> notable-top-25%. As stated in our emails, we are disassociating presentation type from AC recommendations